# Real-World Unsupervised Models Generalize to Predict Brain Responses to Out-of-Distribution Stimuli

Chenggang Chen [1]  Zhiyu Yang [1]  Xiaoqin Wang [1]

## Abstract

Deep neural networks currently provide the leading quantitative models of neural responses in sensory systems. However, these networks remain implausible as models of sensory development, largely because they rely on supervised training with label efficiency far exceeding that of biological learning. Furthermore, these models are typically trained on manually curated datasets that lack the statistical properties of the natural environments to which the brain is exposed. Here, we demonstrate that models trained with unsupervised objectives on real-world data significantly outperform supervised models in predicting brain responses across both human auditory and visual cortex. We show that this performance advantage is not driven by network architecture or dataset size, but rather by the data distribution. Crucially, we find that unsupervised models trained on real-world data exhibit remarkable out-of-distribution generalization: a model trained exclusively on Mandarin speech accurately predicts English-driven brain responses, and a model trained on infant head-cam footage predicts adult visual responses to curated object images. Together, our results illustrate how deep neural networks can be used to reveal the real-world statistics that shape neural representations in the brain.

## 1. Introduction

The rapid proliferation of high-performing deep neural networks (DNNs) in vision and audition has established them as promising computational models for sensory systems. These models offer a new framework to study how different inductive biases shape the emergent alignment between artificial and biological representations (Yamins & DiCarlo,

2016). We posit that the gradients of sensory learning in both systems are governed by three fundamental components: **architecture, objective, and training data.**

Historically, efforts to align DNNs with the brain focused heavily on optimizing **architecture**. Previous studies often relied on massive hyperparameter searches over thousands of candidate models or incorporated computationally expensive, hand-crafted front-ends (e.g., cochlear models) to mimic biological constraints (Yamins et al., 2014; Kell et al., 2018). However, with the advent of the Transformer architecture and the scaling of self-supervised learning, extensive architectural tuning has become less critical. General-purpose engineering models—such as SimCLR in vision and HuBERT in audition—can now be applied end-to-end to effectively model sensory systems (Zhuang et al., 2021; Li et al., 2023), rendering the search for specialized, hypothesis-driven architectures increasingly unnecessary.

Accordingly, the focus has shifted toward the learning **objective**. There is a growing consensus that self-supervised (or unsupervised) learning is superior to supervised learning, not only for its engineering performance but for its biological plausibility. Supervised training, which necessitates massive labeled datasets, is an unlikely candidate for biological learning. Human and animal perception is adapted to real-world environments where labels are scarce or non-existent. For instance, human infants acquire phonetic and linguistic categories purely through the statistics of speech sounds in their native languages, without explicit supervision (Bergelson & Swingley, 2012).

While the field has rigorously explored algorithms and objectives, the third pillar—**training data**—remains the critical missing link. Standard practice relies on curated, "sterile" datasets like ImageNet or LibriSpeech, which present a balanced and noiseless view of the world. In contrast, the biological brain is forged in a chaotic, real-world environment. We argue that the statistical "imperfections" of natural stimuli—specifically extreme variations in density, contrast, and quality—are not obstacles to learning, but the very catalysts that drive robust perception. Essentially, the sensory system is strengthened by the adversity of its input.

First, the brain exploits the challenge of skewed density. Un-

[1]Department of BME, Johns Hopkins University. Correspondence to: Chenggang Chen <cheng-gang.chen@jhu.edu>.

*Proceedings of the 43rd International Conference on Machine Learning*, Seoul, South Korea. PMLR 306, 2026. Copyright 2026 by the author(s).

like the artificial balance of machine learning dataset, natural experience is heavily long-tailed; infants, for example, view the same few faces and objects thousands of times. Far from being inefficient, this massive redundancy forces neural circuits to develop specialized adaptation mechanisms—such as repetition suppression and facilitation (Ulanovsky et al., 2003; Chen et al., 2025). Second, the brain builds resilience through variable contrast. The natural world fluctuates wildly between silence and roar, darkness and glare. To survive this high dynamic range, the nervous system is forced to implement contrast gain control (Heeger, 1992; Rabinowitz et al., 2011). This adaptive mechanism ensures that representations remain stable and informative regardless of input intensity, a form of invariance that standard DNNs trained on normalized data rarely need to develop. Finally, biological representations are defined by their resistance to degraded quality. Real-world signals are rarely clean; they are corrupted by background noise, reverberation, and occlusion. This environmental "hostility" forces the sensory cortices to extract the invariant structure of the signal (Mesgarani et al., 2014) and achieve a "cocktail party" robustness that clean training data simply cannot induce.

Consequently, we hypothesize that achieving high fidelity in brain prediction requires a model to mirror the constraints of biological learning. We propose that an optimal model should be relatively architecture-agnostic but strongly constrained by an unsupervised objective and trained on data that reflects the statistical properties of the real world. In this study, we validate this hypothesis, demonstrating that models aligning with these biological principles consistently yield the most accurate predictions of neural responses.

## 2. Related work

**Supervised models of visual and auditory cortex.** Supervised Deep Neural Networks (DNNs) have been extensively utilized to model primate vision (Cadieu et al., 2014; Khaligh-Razavi & Kriegeskorte, 2014; Yamins et al., 2014; Kubilius et al., 2019; Storrs et al., 2021) and the primate auditory cortex, employing both fMRI (Güçlü et al., 2016; Kell et al., 2018; Millet & King, 2021; Giordano et al., 2023; Tuckute et al., 2023) and intracranial recordings (Keshishian et al., 2020; Mischler et al., 2023; Ahmed et al., 2025; Norman-Haignere et al., 2025; Rupp et al., 2025). These architectures range from purely convolutional networks to hybrid models incorporating recurrent layers (Kubilius et al., 2019) or Transformer blocks (Ahmed et al., 2025). While effective, the majority of these models rely on datasets manually curated for engineering applications rather than biological realism. A notable exception is the work of Mehrer et al. (2021), who introduced Ecoset—a dataset with categories selected for their linguistic frequency and concreteness to better reflect physically salient objects

in the real world. Although they demonstrated that supervised training on Ecoset significantly improves predictions of higher-level visual cortex representations, the reliance on millions of explicit labels remains fundamentally inconsistent with the constraints of biological learning.

**Unsupervised models of visual and auditory cortex.** Recently, unsupervised models have been evaluated as candidate models of the visual cortex without the need for task-specific fine-tuning (Konkle & Alvarez, 2022; Prince et al., 2024). However, the majority of these studies rely on pretraining with ImageNet (Deng et al., 2009), a highly curated dataset comprising 1,000 balanced classes where each image typically centers on a single subject. In the auditory domain, unsupervised modeling has been less explored than supervised approaches. The primary unsupervised architectures investigated thus far are Wav2Vec 2.0 (Baevski et al., 2020) and HuBERT (Hsu et al., 2021). These models are typically pretrained on high-quality, curated speech corpora (Millet et al., 2022; Vaidya et al., 2022; Li et al., 2023), such as LibriSpeech (English) (Panayotov et al., 2015) and MagicData (Mandarin) (Yang et al., 2022). Much like ImageNet, these datasets are derived from audiobooks and are structured primarily to support downstream engineering tasks like Automatic Speech Recognition (ASR). While recent work by Conwell et al. (2024) suggests that training data distribution—rather than architecture or objective—is the dominant factor in neural predictivity, the datasets they examined (faces, objects, places) remain heavily curated and distinct from the statistics of real world (Fig. 1).

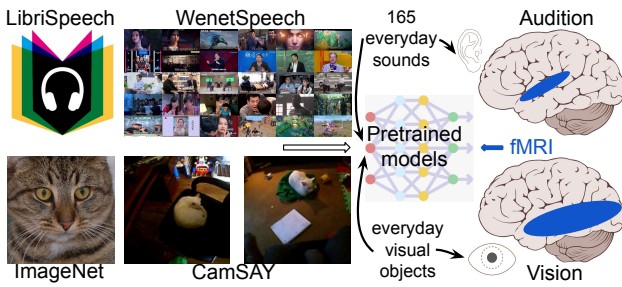

*Figure 1.* Overview of this study. Audio and vision models were pretrained on either clean (LibriSpeech, ImageNet) or real-world (WenetSpeech, CamSAY) datasets. The same set of everyday sounds and visual objects was passed through these pretrained models to extract neural activations and presented to human participants to record fMRI responses in the auditory and visual cortices.

**Out-of-distribution generalization.** Previous studies have consistently demonstrated that DNNs suffer from reduced accuracy when predicting brain responses to out-of-distribution stimuli. For instance, Li et al. (2023) and Millet et al. (2022) investigated this language-specificity by feeding English, Mandarin, and French speech into models pretrained on either matching or mismatched languages. Both

studies found that neural predictivity in the higher-order auditory cortex is maximized when the model's training language matches the listener's native language, indicating a failure to generalize fully across linguistic domains. In the visual domain, Zhuang et al. (2021) trained models solely on developmentally realistic video data collected from infant head-mounted cameras. However, compared to standard models trained on the curated ImageNet dataset, these real-world trained models exhibited lower predictivity of visual cortical responses in macaque monkeys. Consequently, it remains an open question whether models trained on uncurated, real-world data can effectively generalize to predict brain responses to out-of-distribution stimuli. Our main contributions are as follows:

Systematic benchmarking of learning paradigms: We provide the first systematic benchmark comparing supervised and unsupervised models across multiple training data regimes. Unlike prior studies that focused exclusively on supervised models (Tuckute et al., 2023) or models trained on curated speech (Li et al., 2023; Millet et al., 2022), our evaluation incorporates real-world datasets for both audition (Zhang et al., 2022) and vision (Sullivan et al., 2021).

Unsupervised models surpass supervised baselines: We demonstrate, for the first time, that unsupervised models trained on real-world data without any fine-tuning can significantly outperform supervised models in predicting neural responses in both the auditory and visual cortices. This contrasts with previous findings, which suggested that unsupervised models could at best close the performance gap with supervised models (Li et al., 2023; Zhuang et al., 2021).

Disentangling data distribution from architecture: We isolate the source of this superior predictivity, revealing that it is driven primarily by the distribution of the training data (real-world vs. curated) rather than model architecture, parameter count, or dataset size. By controlling for these variables, we resolve ambiguities present in prior literature regarding the specific factors driving neural alignment.

Out-of-distribution (OOD) generalization: We are the first to show that models trained on the real-world data can generalize to predict brain responses to OOD stimuli never encountered during training. This indicates that unsupervised learning on real-world data offers a strong and biologically plausible computational theory of sensory representations and learning in the brain.

## 3. Models, datasets, and metrics

Building on the benchmark established by Tuckute et al. (2023), we analyzed activations from a suite of 19 audio models (Table 1). There are ten in-house models using two architectures, CochCNN9 and CochResNet50, which modify the standard ResNet50 (He et al., 2016) and CNN archi-

tectures to accept cochleagram inputs. These models were trained on a constructed Word-Speaker-Noise dataset (128 hours) (Feather et al., 2019), which combined speech clips (from Wall Street Journal and Spoken Wikipedia Corpora) with environmental sounds (from AudioSet) (Gemmeke et al., 2017) to enable three distinct training objectives: word recognition, speaker identification, and environmental sound classification (via AudioSet background labels), along with a fourth multitask objective combining all three. Separately, to investigate music-specific representations, models were trained on the Million Song Dataset (Bertin-Mahieux et al., 2011) to perform a music genre classification task.

We retained four high-performing external models from the original study. AST (Gong et al., 2021) is trained on the AudioSet dataset, which comprises 2 million 10-second clips (5,800 hours), to classify the 527 AudioSet sound event classes. VGGish (Hershey et al., 2017) is trained on the YouTube-8M dataset, comprising 8 million videos (500,000 hours), to classify 30,871 video-level topic annotations. S2T (Wang et al., 2020) is a supervised encoder-decoder Transformer trained on the LibriSpeech-960h dataset to decode a 10,000-token unigram vocabulary. In contrast, Wav2Vec2FT (Baevski et al., 2020) was pretrained via unsupervised learning on LibriSpeech-960h and subsequently fine-tuned on the same dataset to classify 32 characters.

We excluded five (DCASE2020, DeepSpeech2, MetricGAN, SepFormer, and VQ-VAE) models that previously demonstrated low predictivity of brain data. In their place, we introduced five unsupervised models to test the efficacy of modern representation learning. Specifically, we utilized the HuBERT and Wav2Vec2 frameworks. These two architectures share an identical backbone—comprising a seven-layer convolutional feature extractor followed by a 12-layer Transformer encoder—but optimize distinct unsupervised objectives: HuBERT employs a masked predictive loss (predicting discrete cluster assignments), whereas Wav2Vec2 utilizes a contrastive loss. To ensure fair comparison with previous baselines, we utilized the "Base" configuration (approx. 95M parameters) for all unsupervised models. As detailed in our results, we find that despite their differing objectives, HuBERT and Wav2Vec2 exhibit comparable performance when controlled for training data.

To disentangle the effects of architecture, dataset scale, and acoustic diversity, we evaluated the five unsupervised models across three distinct datasets. First, we utilized the LibriSpeech-960h (LS) corpus, which was also used to train the supervised S2T and Wav2Vec2FT baselines. We employed HuBERTLS and Wav2Vec2LS as control models to enable direct comparisons with previous work while holding the training data constant. Second, to test whether dataset size plays a major role, we included HuBERTcore (Hagiwara, 2023). This model is pre-trained on a smaller,

*Table 1.* Summary of model architectures, outputs, and training datasets. The top five rows represent unsupervised learning models newly added to the benchmark. HuBERT-based and Wav2Vec2-based models are denoted by diamond (♦) and circle (●) markers, respectively.

| Model Name | Brief Description | Model Output | Training Dataset |
|---|---|---|---|
| HuBERTspeech ♦ | Transformer arch. for unsupervised learning | Latent repre. (768) | WenetSpeech |
| Wav2Vec2speech ● | Transformer arch. for unsupervised learning | Latent repre. (768) | WenetSpeech |
| HuBERTLS ♦ | Transformer arch. for unsupervised learning | Latent repre. (768) | LibriSpeech-960h |
| Wav2Vec2LS ● | Transformer arch. for unsupervised learning | Latent repre. (768) | LibriSpeech-960h |
| HuBERTcore ♦ | Transformer arch. for unsupervised learning | Latent repre. (768) | FSD50k+AudioSet |
| AST (Audio Spec. Transformer) | Transformer arch. for audio classifi. | AudioSet label (527) | AudioSet |
| VGGish | Convolutional arch. for audio classifi. | Video label (30,871) | YouTube-8M |
| S2T (Speech-to-Text) | Transformer arch. for auto. speech recog. | Words (10,000) | LibriSpeech-960h |
| Wav2Vec2FT ● | Transformer arch. for auto. speech recog. | Characters (32) | LibriSpeech-960h |
| CochCNN9/ResNet50 Word | Convolutional arch. for word recog. | Word label (794) | Word-Speaker-Noise |
| CochCNN9/ResNet50 Speaker | Conv. arch. for speaker recog. | Speaker label (433) | Word-Speaker-Noise |
| CochCNN9/ResNet50 AudioSet | Conv. arch. for auditory event recog. | AudioSet label (517) | Word-Speaker-Noise |
| CochCNN9/ResNet50 MultiTask | Conv. arch. for word/speaker/AS recog. | W/S/AS label | Word-Speaker-Noise |
| CochCNN9/ResNet50 Genre | Conv. arch. for music genre classifi. | Genre label (41) | Million Song |

diverse dataset (153 hours) comprising FSD50K (Fonseca et al., 2021) and the balanced subset of AudioSet. Finally, we incorporated models trained on WenetSpeech (Zhang et al., 2022), a large-scale corpus containing 10,000 hours of Mandarin speech. A key advantage of this dataset relative to LibriSpeech is its high variability; it covers a broad spectrum of speaking styles, scenarios, domains, topics, and noisy conditions, allowing us to assess robustness to out-of-distribution, real-world stimuli (see Methods).

Tuckute et al. (2023) evaluated their models using two independent fMRI datasets, both of which presented the same set of 165 two-second natural sounds to human listeners. These stimuli comprise 158 everyday sounds, rigorously screened via behavioral experiments for high human recognizability and daily exposure frequency, plus seven foreign speech clips. The first dataset (NH2015; Norman-Haignere et al., 2015) includes data from 8 participants with moderate musical experience. The second dataset (B2021; Boebinger et al., 2021) comprises data from 20 distinct participants: 10 with minimal musical training and 10 with extensive training. To ensure consistent engagement, both datasets required participants to perform a sound intensity discrimination task during scanning, pressing a button upon detecting a quieter target sound embedded within the stimulus blocks.

Extensive research has sought to establish robust methods for comparing learned representations in DNNs with brain activity. Two primary frameworks have emerged. The first utilizes encoding models, which learn a mapping from model features to brain activity on a training set to predict responses in held-out data ($r^2$) (Naselaris et al., 2011).

The second approach, Representational Similarity Analysis (RSA), abstracts away from direct feature-to-voxel mapping to compare the geometric structure of representational spaces via dissimilarity matrices (Kriegeskorte et al., 2008). In this work, we leveraged both encoding models (Figs. 2, 5, 6) and RSA (Figs. 3, 4) to analyze auditory representations. For the visual brain, we employed encoding models and the Brain-Score benchmark (Schrimpf et al., 2018) (Figs. 7, 8).

## 4. Results

### 4.1. Dataset distribution plays a pivotal role in explaining auditory brain responses

We found that two of the five newly introduced unsupervised learning models achieved the top-1 and top-2 performance in predicting auditory responses within the NH2015 dataset (Fig. 2). Importantly, while the gap between the top-1 (0.729) and top-2 (0.726) models in the previous benchmark was only 0.4%, our additions represent a substantial leap in performance: HuBERTspeech (0.773) and Wav2Vec2speech (0.743) outperformed the previous state-of-the-art (SOTA) by 6% and 1.9%, respectively. Furthermore, HuBERT-Speech achieved higher variance explained ($r^2$) than the previous best supervised model (CochResNet50-MultiTask) in 8 out of 10 participants. These results were consistent in the B2021 dataset, where HuBERTspeech again ranked first, with Wav2Vec2speech following closely behind the previous top-two models. Collectively, these findings demonstrate that audio models pretrained on large-scale, real-world data achieve SOTA accu-

racy in predicting human brain activity.

We found that the unsupervised HuBERTcore, despite its smaller training dataset (153 hours), outperformed 9 and 6 supervised models on the NH2015 and B2021 datasets, respectively. For example, in the B2021 dataset with the RSA metric (see below), HuBERTcore outperforms HuBERTLS by 15.5% (Table 2). This suggests that the data distribution (real-world vs. curated) contributes more significantly to brain prediction than dataset size (153 vs. 960 hours).

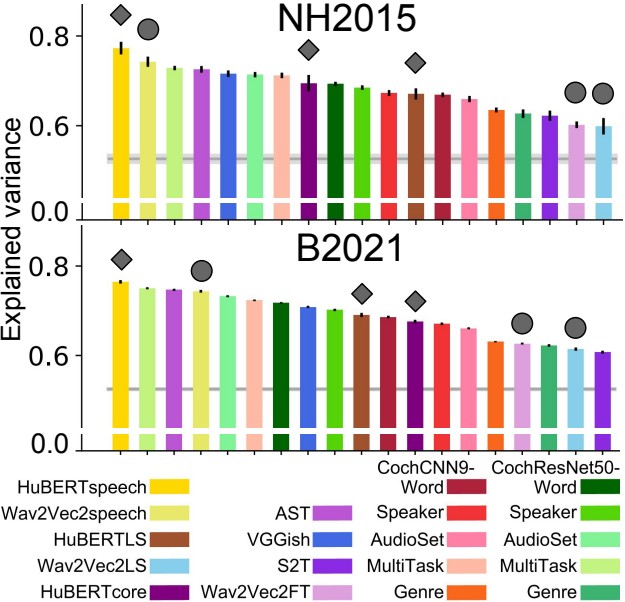

*Figure 2.* Brain prediction accuracy. Explained variance ($r^2$) was assessed for each voxel using regression. The plot displays the aggregated median variance explained for the best-predicting stage of each model, selected using held-out data. The gray line indicates the performance of the SpectroTemporal baseline model.

Although Wav2Vec2speech achieved top-2 and top-4 rankings across the two benchmarks, Wav2Vec2FT (LibriSpeech pretrained and fine-tuned) and Wav2Vec2LS (LibriSpeech pretrained only) ranked significantly lower in both datasets. This suggests that the training dataset plays a key role in predicting brain responses. Furthermore, fine-tuning the pretrained models yielded no improvement (NH2015) or even deteriorated (B2021) brain prediction ability.

To determine whether naturalistic training captures novel biological properties or simply sacrifices clean-sound processing for noise tolerance, we examined how these models perform out-of-distribution on "clean" Speech Commands (Warden, 2018) datasets. HuBERTLS pretrained on the curated 960-hour LibriSpeech dataset, achieves a 96.3% accuracy under the SUPERB benchmark (Yang et al., 2021). In contrast, HuBERTcore (Hagiwara, 2023) pretrained on 360 hours of uncurated naturalistic and animal sounds achieves a 96.4% accuracy on the exact same "clean" dataset. The fact that naturalistically trained unsupervised

models mimic this behavior—maintaining acoustic accuracy on clean datasets while exhibiting higher alignment with fMRI responses—suggests these models capture the inherent environmental invariance of the auditory cortex.

### 4.2. Unsupervised training on real-world data yields the strongest representational similarity to auditory cortical responses

We next turned our attention to a complementary metric: RSA. Unlike encoding models that focus on individual voxels, this metric constructs Representational Dissimilarity Matrices (RDMs) for both model features and brain responses, quantifying their alignment using Spearman correlation. Despite employing a fundamentally different metric, we observed highly consistent findings: HuBERTspeech and Wav2Vec2speech achieved the top-1 and top-3 rankings in the NH2015 dataset, and the top-2 spots in B2021 (Fig. 3). In the NH2015 dataset, the top three models (including CochResNet50-MultiTask; scores: 0.506, 0.506, 0.502) yielded comparable performance but outperformed the fourth-ranked model (0.491) by at least 2.2%. In the B2021 dataset, the top two unsupervised models (0.488, 0.473) surpassed the previous state-of-the-art (0.461) by 5.9% and 2.6%, respectively, whereas the margin between the previous top-1 and top-2 (0.458) was only 0.66%.

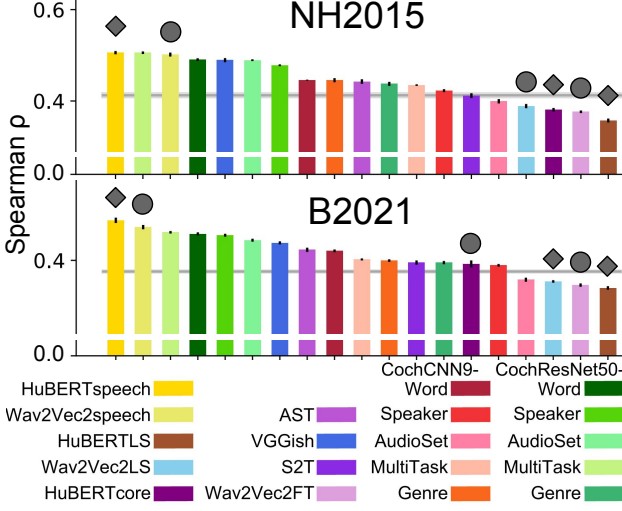

*Figure 3.* Representational similarity between fMRI responses and model activations. The five newly introduced models are highlighted with diamond (♦) and circle (•) markers. The gray line denotes the representational similarity (Spearman correlation) achieved by the SpectroTemporal baseline model.

Although the unsupervised HuBERT and Wav2Vec2 models pretrained on real-world data achieved the highest rankings, they fell to nearly the lowest rankings when pretrained on manually curated datasets (LibriSpeech). The sole exception was HuBERTcore in the B2021 dataset, which outperformed only two supervised models. Taken together, these

findings demonstrate that unsupervised models pretrained on real-world data—unlike those trained on hand-curated datasets—exhibit the highest representational similarity to brain responses without the need for fine-tuning.

### 4.3. HuBERTspeech mirrors the topography of brain responses to sound categories

Both fMRI datasets used the same 165 natural sounds, each two seconds in duration. These stimuli were selected from an initial pool of 280 common sounds based on recognizability and frequency of daily occurrence, as validated by Amazon Mechanical Turk surveys. Those 165 daily sounds include 11 categories (Fig. 4, 4 colored boxes), organized into music (blue, 24 instrumental, 11 vocal), speech (green, 10 English, 7 foreign), and non-speech vocalizations (purple, 13 human, 10 animal). The remaining sounds capture distinct sources (red): human non-vocal (15), animal non-vocal (5), nature (4), mechanical (39), and environmental sounds (27). Notice that vocal music could be categorized into either music and speech categories.

We first examined the RDMs of the fMRI data to visualize the topography of brain responses to different sound categories (Fig. 4, top). Both neural datasets display very similar representational structures. The strongest similarity appears as a distinct white block comprising the 10 English (dark green) and 7 foreign (light green) speech sounds. Vocal music (cyan) acts as a bridge, forming clusters with both the 17 speech sounds and the 24 instrumental music sounds (blue). Conversely, the representations of speech and instrumental music are highly distinct from one another. The 23 non-speech vocal sounds form a weakly correlated cluster that separates them from the other categories. Finally, the remaining environmental, mechanical, and non-vocal sounds formed a single, broad, and loosely correlated cluster. Together, the representational topography observed in the fMRI RDMs closely aligns with the perceptual categories defined by human participants.

We next examined the RDMs derived from the HuBERT-speech model, which achieved the highest performance regardless of model architecture and dataset type (Fig. 4, middle). Visual inspection reveals that the overall representational topology in HuBERTspeech closely matches the human fMRI data across all four major sound categories. This alignment is particularly striking for the music and speech categories (blue and green dashed boxes), where the model replicates the strong within-category similarity (bright blocks) and the distinct separation between instrumental music and speech observed in the neural data. Furthermore, HuBERTspeech preserves the hierarchical structure seen in the brain: vocal music acts as a connective bridge sharing features with both speech and instrumental music, non-speech vocal sounds (purple) form as a unique

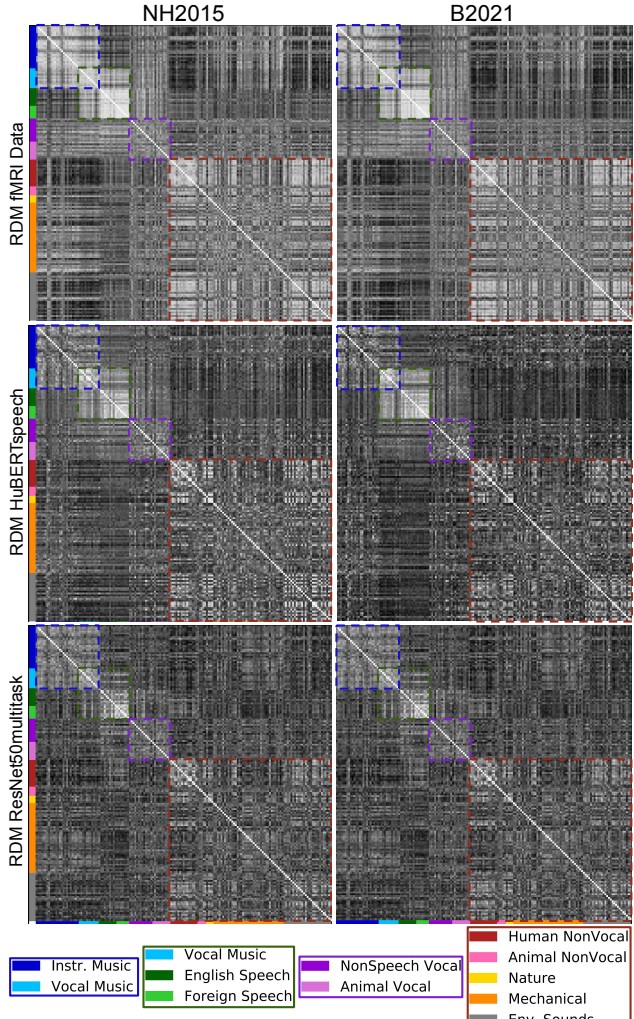

*Figure 4.* RDMs of the auditory cortex and models. Rows display the neural and model RDMs, while columns correspond to the two fMRI datasets. The color scale represents dissimilarity values ranging from 0 to 1: white indicates low dissimilarity (similar representations), while black indicates high dissimilarity.

category, while the environmental and mechanical sounds (red) form a large, broad cluster that is clearly distinct from the vocal and musical domains.

Finally, we examined the CochResNet50-MultiTask, the top-performing supervised model (Fig. 4, bottom). Most notably, vocal music (cyan) clusters exclusively with instrumental music (blue) while maintaining a strong dissimilarity from speech (green), failing to capture the shared 'vocal' features observed in the brain. Furthermore, the model struggles to recover the broader categorical structure: the non-speech vocalizations (purple) do not form a coherent cluster, and the remaining environmental and mechanical sounds (red) exhibit a weak representations.

To compare the RDMs between HuBERTspeech and

CochResNet50-MultiTask quantitatively, we used Centralized Kernel Alignment (CKA) and Mutual k-Nearest Neighbors (MkNN). CKA is a robust metric used to evaluate global representational similarity (Kornblith et al., 2019). We applied CKA to the RDMs by double-centering the distance matrices and calculating their normalized Hilbert-Schmidt Independence Criterion (HSIC). On the NH2015 dataset, HuBERTspeech achieved a higher CKA score (0.7182) compared to ResNet50multitask (0.7012). This superior global alignment was replicated on the B2021 dataset (HuBERTspeech: 0.7331 vs. ResNet50multitask: 0.7013). Figure 17 shows the quantitative comparison of RDMs using MkNN, which evaluates local topological geometry and neighborhood preservation. At small neighborhood sizes ($k < 40$ for NH2015; $k < 10$ for B2021), ResNet50multitask shares a higher proportion of immediate neighbors with the neural data, indicating stronger alignment with fine-grained, low-level acoustic clustering. Conversely, as the neighborhood expands, the curves intersect, and HuBERTspeech demonstrates greater overlap.

In summary, ResNet50multitask better captures fine-grained acoustic neighborhoods (small $k$ in MkNN), whereas HuBERTspeech better captures the global geometry and broad categorical structure (large $k$ in MkNN and higher CKA).

### 4.4. HuBERTspeech best predicts the speech-selective neural component

To examine model predictions for specific tuning properties of the auditory cortex, Tuckute et al. (2023) utilized a previously derived set of response components. Norman-Haignere et al. (2015) demonstrated that cortical voxel responses to natural sounds can be modeled as a linear combination of six response components via voxel component decomposition. These six components are interpreted as capturing the tuning properties of underlying neural populations. Two components are primarily driven by audio frequency content (tonotopy), while two others reflect tuning to spectral and temporal modulation frequencies. The remaining two components exhibit high selectivity for speech and music, respectively. Notably, these components show distinct anatomical distributions, with speech and music selectivity being most prominent in the lateral and anterior regions of the non-primary auditory cortex (Fig. 11, 12).

Figure 5 displays the relationship between the actual (x-axis) and predicted (y-axis) responses for the speech component. The predicted responses from HuBERTspeech closely matched the actual neural data, explaining 88% of the variance. High responses were elicited almost exclusively by sounds categorized as "English speech" (dark green) and "Foreign speech" (light green), followed by vocal music (cyan), which contains speech content via lyrics. Responses to non-speech vocalizations (human or animal) were ele-

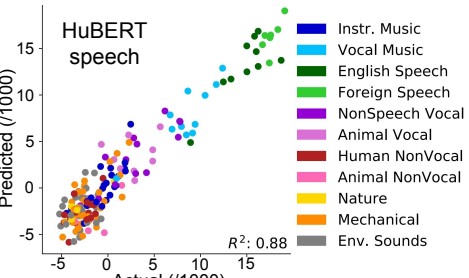

*Figure 5.* Comparison of actual fMRI responses (x-axis) and HuBERTspeech predicted responses (y-axis) for the speech component across the 165 natural sounds.

vated compared to non-vocal sounds but remained substantially lower than those to speech. Notably, foreign speech elicited responses at least as high as English speech, a pattern that persisted even after excluding foreign languages familiar to the participants. These results suggest that the speech component responds selectively to speech structure itself, independent of linguistic comprehension. This finding plausibly explains why models like HuBERTspeech and Wav2Vec2speech—despite being pretrained on Wenet-Speech, a Mandarin corpus—can accurately predict brain responses in native English speakers.

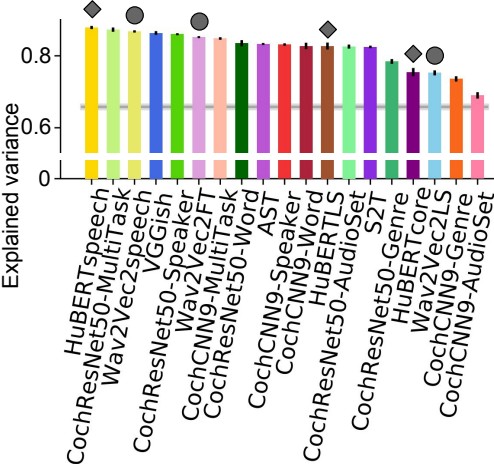

*Figure 6.* A benchmark for the responses of the speech component across 19 models (similar to Fig. 2). Five newly added models and one previous supervised model (Wav2Vec2FT) are highlighted with diamond (♦) and circle (•) markers. The gray line shows variance explained by the SpectroTemporal baseline model.

Among the 19 evaluated models, the unsupervised HuBERTspeech and Wav2Vec2speech ranked first and third on the NH2015 dataset (Fig. 6). This is consistent with previous results analyzing the full neural response (Fig. 2, 3). A notable difference here is the performance of the fine-tuned Wav2Vec2FT, which ranked 6th, suggesting that speech fine-tuning improved the model's predictive ability relative to its baseline. In contrast, models trained on data with lim-

ited speech access (e.g., HuBERTcore) or manually curated datasets without fine-tuning (e.g., Wav2Vec2LS) exhibited lower predictive accuracy. Collectively, the analysis of the speech component consistently demonstrates that unsupervised models pretrained on real-world data are the best predictors of auditory brain responses.

### 4.5. Unsupervised learning from child-perspective video yields superior brain prediction

After evaluating 19 unsupervised and supervised models in two fMRI datasets from the human auditory cortex, we now turn our attention to the human high-level visual cortex. Although unsupervised learning models can acquire robust visual representations without labels, the standard training substrate—ImageNet—diverges significantly from the visual diet of human infants. ImageNet is characterized by category-balanced data containing distinct object instances, often captured from clean, stereotypical angles. In contrast, infants encounter a much smaller set of object instances viewed under highly variable and noisy conditions (Smith & Slone, 2017). To better approximate this developmental experience, we utilized the SAYCam dataset (Sullivan et al., 2021), a large, naturalistic, longitudinal corpus collected from head-mounted cameras on infants aged 6–32 months (recorded 2 hours per week over 2.5 years). We evaluated ResNeXt architectures trained on video streams from individual infants (S, A, and Y) as well as the combined cohort (Orhan et al., 2020). For brevity, we refer to these models as camS, camA, camY, and camSAY, respectively.

The Brain-Score benchmark (Schrimpf et al., 2018) currently includes two human visual cortex fMRI datasets. We first utilized the dataset from (Coggan & Tong, 2023), who measured 7T fMRI responses in 10 human subjects. The stimuli were derived from a study by (Bao et al., 2020) and also included a classic category set consisting of 80 images of faces, bodies, houses, and common objects (Fig. 7). We benchmarked the four newly added models against 493 existing entries on the 'tong.Coggan2024_fMRI.IT' dataset in Brain-Score. The four models, pretrained on real-world data, achieved Brain-Scores of 0.438, 0.4167, 0.4137, and 0.393, outperforming the benchmark's previous leader (swin-small, 0.267) by at least 47%. Collectively, these findings demonstrate that models trained on unsupervised, head-mounted camera footage are the superior predictors of neural responses in the human high-level visual cortex.

We also employed the dataset from Bracci et al. (2019), which contains 3T fMRI responses recorded from 17 human subjects. The stimulus set consists of nine triads, each comprising an animal, a typical object, and a "lookalike" object resembling the animal (Fig. 8a, b). To control for architectural confounds, we benchmarked all models using the ResNeXt backbone (Xie et al., 2017). The camSAY models

utilize the ResNeXt-50-32x4d architecture, characterized by a depth of 50 layers, a cardinality of 32 (groups of convolutions), and a bottleneck width of 4 channels. Accordingly, we tested the identical ResNeXt-50-32x4d architecture pretrained on ImageNet as a baseline. Furthermore, given that scaling depth, cardinality, and width is known to improve model performance, we additionally evaluated the ResNeXt-101-32x8d and ResNeXt-101-64x4d variants. These larger models feature a depth of 101 layers, with cardinalities of 32 and 64, and channel widths of 8 and 4, respectively.

The top-1 accuracies of ResNeXt-101-64x4d, 101-32x8d, and 50-32x4d on ImageNet-1K are 83.2%, 79.3%, and 77.6%, respectively. Consistent with this hierarchy, the Brain-Scores for these three models also decreased monotonically: 0.21, 0.196, and 0.169 (Fig. 8c). However, even the best-performing ImageNet model (ResNeXt-101-64x4d) yielded a Brain-Score 18.6% lower than the *lowest*-performing ResNeXt trained on the camSAY dataset (scores: 0.259, 0.257, 0.253, and 0.249). Together, these results indicate that the superior performance of the camSAY-ResNeXt models is attributable to the nature of the pretraining data rather than the capacity of the ResNeXt architecture.

To further isolate the effect of the learning objective, we expanded this comparison to include state-of-the-art unsupervised models, specifically DINO variants (vit_base_dinov2-lvd142m, dinov2-lvd142m, and dinov3-lvd1689m). The fMRI benchmark stimuli consist of highly curated, adult-centric images. Because the DINO models were optimized on 142 million to 1.6 billion similarly curated, diverse web images (Oquab et al., 2023; Siméoni et al., 2025), they evaluate these benchmark stimuli In-Distribution (ID). In contrast, our baby-vision models are trained on egocentric footage bounded by the developmental and physical constraints of an infant's daily life. This biologically plausible data possesses a highly skewed, long-tailed distribution dominated by uncurated, often distorted close-ups of a limited set of household objects and caregivers. Canonical photos of distant vehicles are entirely foreign to this regime. Consequently, our baby-vision models are forced to predict brain responses to stimuli that are Out-of-Distribution (OOD). This dynamic perfectly mirrors our auditory findings. Just as native English speech is a "foreign" OOD stimulus to our models pretrained exclusively on Mandarin, curated ImageNet-style photographs are "foreign" OOD stimuli to our baby-vision models. With this context, the performance of vision models becomes striking. Despite a disadvantage in data volume, an older architecture, and the penalty of evaluating on OOD stimuli, our unsupervised baby-vision models achieve Brain-Scores (0.249 to 0.259) that are comparable with the ID-evaluated DINO models (0.251 to 0.262).

Together, our audio and vision results converge on a unified conclusion: unsupervised models trained on the statistics of

naturalistic environments (whether baby-vision or foreign speech) learn representations so robust and universal that they can generalize to OOD stimuli as effectively as massive, brute-force models predicting ID stimuli.

## 5. Discussion

In this study, we evaluated 5 unsupervised and 14 supervised audio models pretrained on both curated and real-world datasets, benchmarking their predictivity against two fMRI datasets of the human auditory cortex. Extending our analysis to vision, we evaluated four unsupervised models trained on egocentric images and compared their performance on Brain-Score against current state-of-the-art benchmarks. Across both sensory modalities, we consistently observed that unsupervised models trained on real-world data are the superior predictors of brain responses.

Note that the two metrics used are limited to predictive or representational alignment, disregarding mechanistic correspondence between models and the brain. In an encoding model, good matching might mean that the model features contain enough information about the stimuli. In RSA, similar representational geometry does not necessarily mean similar learned features. To complement them, we take two strategies. One is to compare the tuning curves of the model units with the fMRI voxel or component. The other is to compare RDMs between model activations and brain responses using CKA and MkNN.

Our quantitative analyses reveal two key insights into model-brain alignment. First, regarding training data, we observe a functional trade-off: while massive, curated datasets (HuBERTLS) can sometimes yield higher raw voxel-wise predictivity, pretraining on a much smaller, uncurated, real-world dataset (HuBERTcore) consistently yields superior global representational alignment (measured by RSA). This suggests that naturalistic data distributions serve as a highly efficient inductive bias for capturing the brain's global topological geometry. Second, regarding learning objectives, our analyses using MkNN and CKA indicate that supervised and self-supervised models optimize for different levels of the auditory processing hierarchy. Supervised models (e.g., CochResNet50-MultiTask) excel at capturing fine-grained, low-level acoustics, whereas self-supervised models (e.g., HuBERTspeech) better capture the global geometry and broad categorical structure of the human auditory cortex.

Perhaps our most striking finding is the capacity of unsupervised models to generalize across vast domain shifts, predicting brain responses to stimuli that are statistically distinct from their training distributions. For instance, the Wenet-Speech model, trained exclusively on Mandarin, achieved state-of-the-art predictivity of cortical responses to English speech—outperforming models trained specifically on cu-

rated English data. Similarly, in the visual domain, models trained on developmentally realistic, egocentric video from infants effectively predicted adult visual cortical responses to standard object images. This presents a challenge to conventional supervised learning perspectives: models trained on naturalistic, yet technically out-of-distribution data (e.g., Mandarin/Infant video) outperformed models trained on curated, domain-matched data (e.g., English/ImageNet).

We propose that the high variability and noise inherent in real-world, uncurated data serve as a critical regularization mechanism. These conditions force unsupervised models to learn robust, invariant, and universal sensory statistics akin to the brain, preventing overfitting to specific linguistic phonemes or semantic categories. This hypothesis is consistent with findings that task-optimized supervised models predict brain responses more accurately when trained in the presence of background noise (Kell et al., 2018; Tuckute et al., 2023). Similarly, DNN models of sound localization have been shown to replicate human-like accuracy only when trained in realistic conditions containing noise and reverberation (Francl & McDermott, 2022); conversely, models trained in unnatural, anechoic environments deviate significantly from human behavior. Collectively, these results suggest that the "ecological validity" of the training data—defined by its inherent skew, variance, and noise—is a more critical determinant for convergence with brain than the balanced, normalized cleanliness of curated datasets.

While data augmentation and self-supervised (or unsupervised) are well-established in machine learning (ML), this study demonstrates their critical value in NeuroAI for bridging artificial and biological intelligence. Specifically, we show that these techniques significantly improve computational models of the brain. Our findings are consistent with modern ML literature (Lin et al., 2024), where both in- and out-of-distribution (ID and OOD) data augmentation improve generalization via implicit regularization.

One future direction is to compare our top-performing models (HuBERT and Wav2Vec2) trained on Mandarin against native Mandarin speakers. We anticipate that matching the training data to the listener's native language will further improve predictivity. This expectation is consistent with prior intracranial (Li et al., 2023) and fMRI (Millet et al., 2022) evidence showing enhanced predictivity to native language.

One limitation of our study is the evaluation metrics we used, including the encoding model, RSA, CKA, and MkNN, where better performance in those metrics does not necessarily imply a mechanistic correspondence with the brain. Although comparing tuning curves between model units and fMRI voxels or components complements those metrics, future work should incorporate additional analyses, such as benchmarking audio embeddings (Chen & Yang, 2025) and correlating them with human behavior (Chen et al., 2026).

## Acknowledgement

This work was supported by National Institutes of Health grants DC003180 and DC021609 (X.W.). We thank four reviewers for their highly constructive comments, which greatly improved this manuscript. The authors declare no conflicts of interest.

## Impact Statement

This paper presents work whose goal is to advance the field of Machine Learning. There are many potential societal consequences of our work, none which we feel must be specifically highlighted here.

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

## A. Supplementary Table

*Table 2.* Quantitative comparison of two self-supervised (or unsupervised) models (HuBERTcore and HuBERTLS) across two datasets (NH2015 and B2021) using two metrics (brain response predictivity and RSA). Because we planned *a priori* to compare two models with the same architecture but different training datasets, we did not correct the $p$-values for multiple comparisons.

| Metric | NH2015 | | B2021 | |
|---|---|---|---|---|
| | **Prediction** | **RSA** | **Prediction** | **RSA** |
| HuBERTcore | 0.6949 | 0.3808 | 0.6758 | 0.3923 |
| HuBERTLS | 0.6708 | 0.3573 | 0.6905 | 0.3396 |
| Difference (%) | 3.6 | 6.6 | -2.1 | 15.5 |
| $T$-statistic | 0.8943 | 3.9561 | 4.0770 | 5.9557 |
| $p$-value | 0.4009 | 0.0055 | 0.0006 | $9.87 \times 10^{-6}$ |

## B. Supplementary Figures

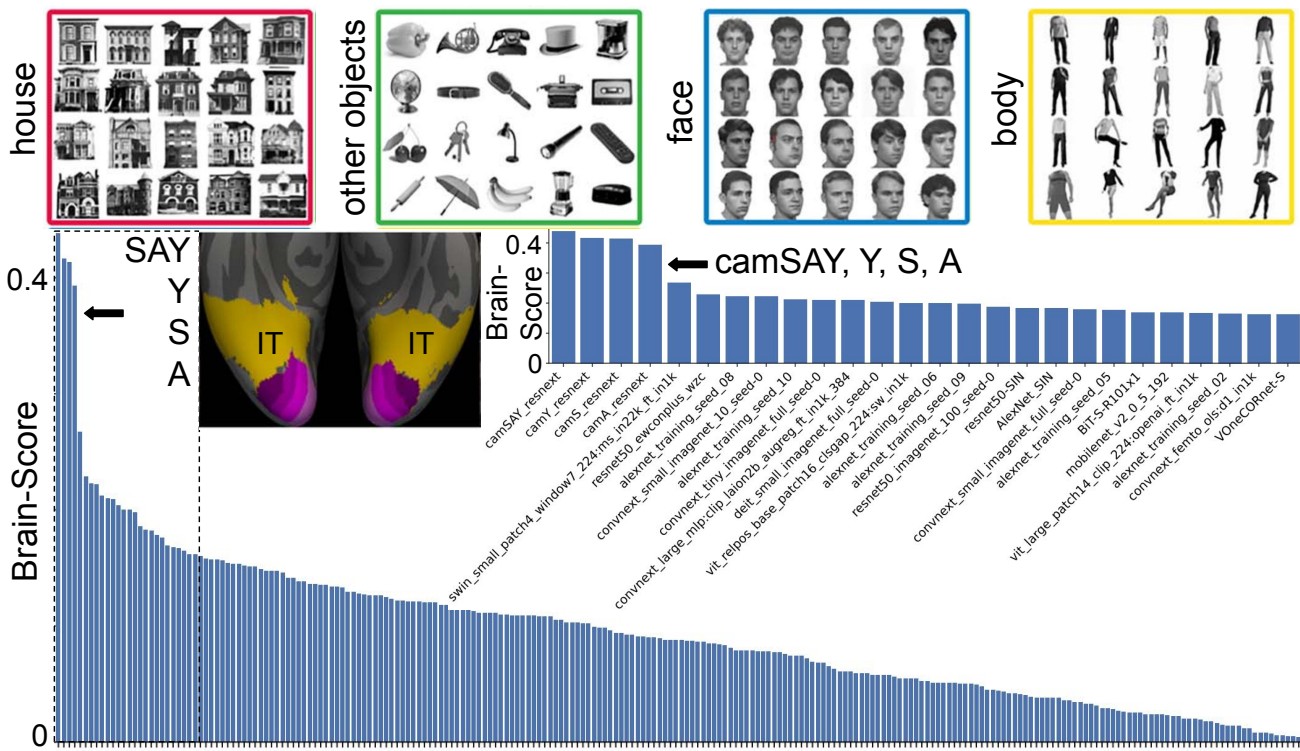

*Figure 7.* Four unsupervised models trained on real-world data occupy the top four positions in the Brain-Score benchmark. Top: Exemplar stimuli (houses, common objects, faces, and bodies) used to functionally localize category-selective regions in the human inferior temporal (IT) cortex. Middle-Left: Atlas-based regions of interest (ROIs) encompassing early visual cortex and IT, projected onto an inflated cortical surface (inferior view). Middle-Right: Brain-Score comparison of the four newly added models against the previous top-22 models. Bottom: Comprehensive Brain-Score ranking for all 497 evaluated models.

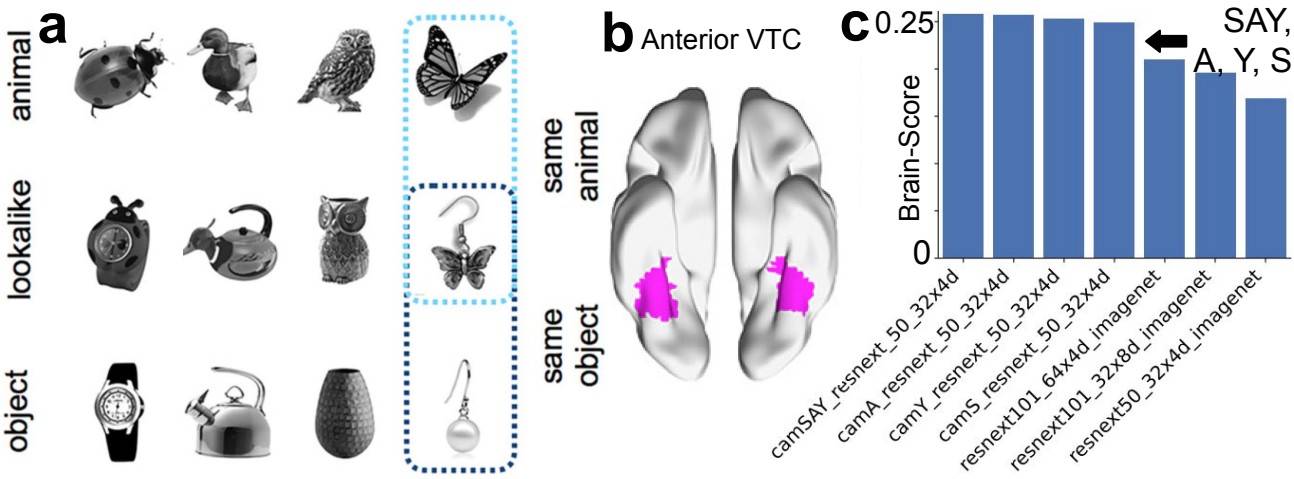

*Figure 8.* Unsupervised models trained on real-world data outperform ImageNet-pretrained models, even those with larger architectures. **a** Example stimuli triads. Each column displays an animal, a typical inanimate object, and a 'lookalike' object that visually resembles the animal. Four of the nine experimental triads are shown. **b** Group-averaged ROIs in the anterior ventral temporal cortex (ant-VTC), visualized on an inflated human brain template. **c** Brain-Score comparison of seven unsupervised ResNeXt models, differentiating those pretrained on real-world video data from those trained on the manually curated ImageNet dataset.

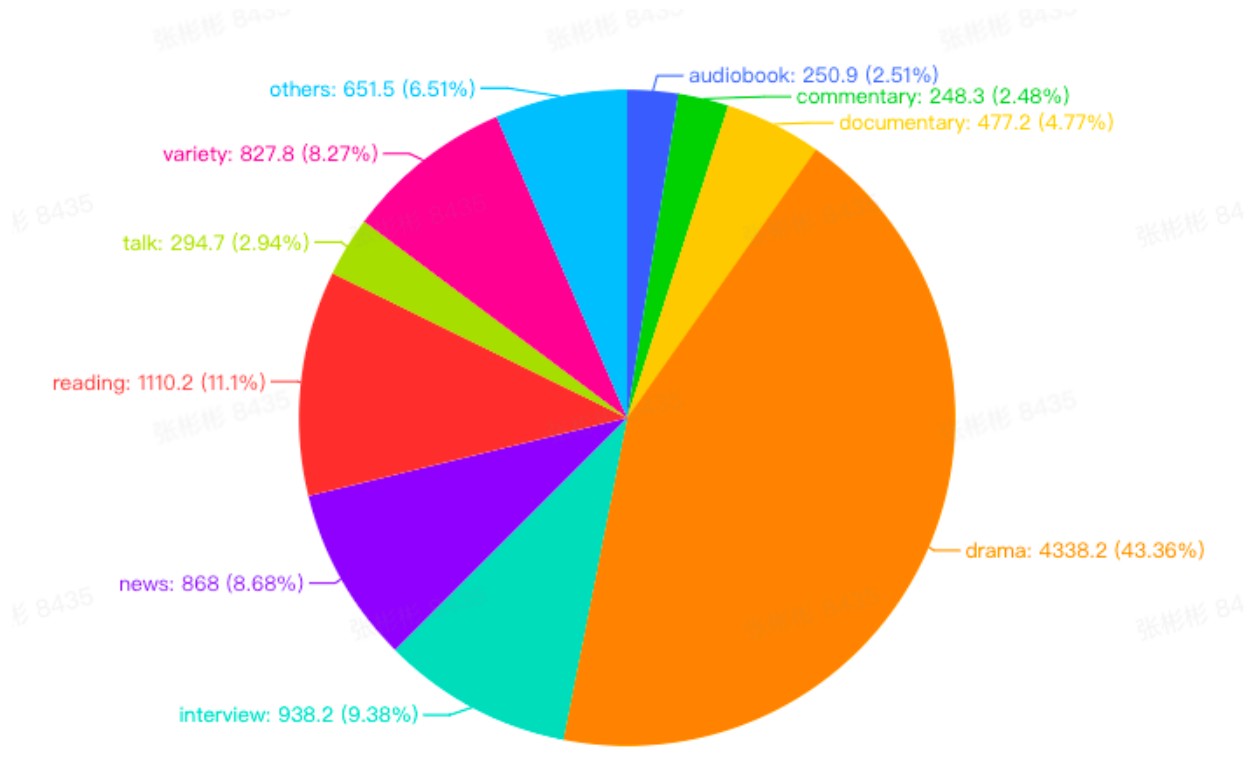

*Figure 9.* Relevant to Table 1. The WenetSpeech (10, 005 hours) can be mainly classified into 10 categories according to speaking styles and spoken scenarios. The figure is copied from https://wenet-e2e.github.io/WenetSpeech/.

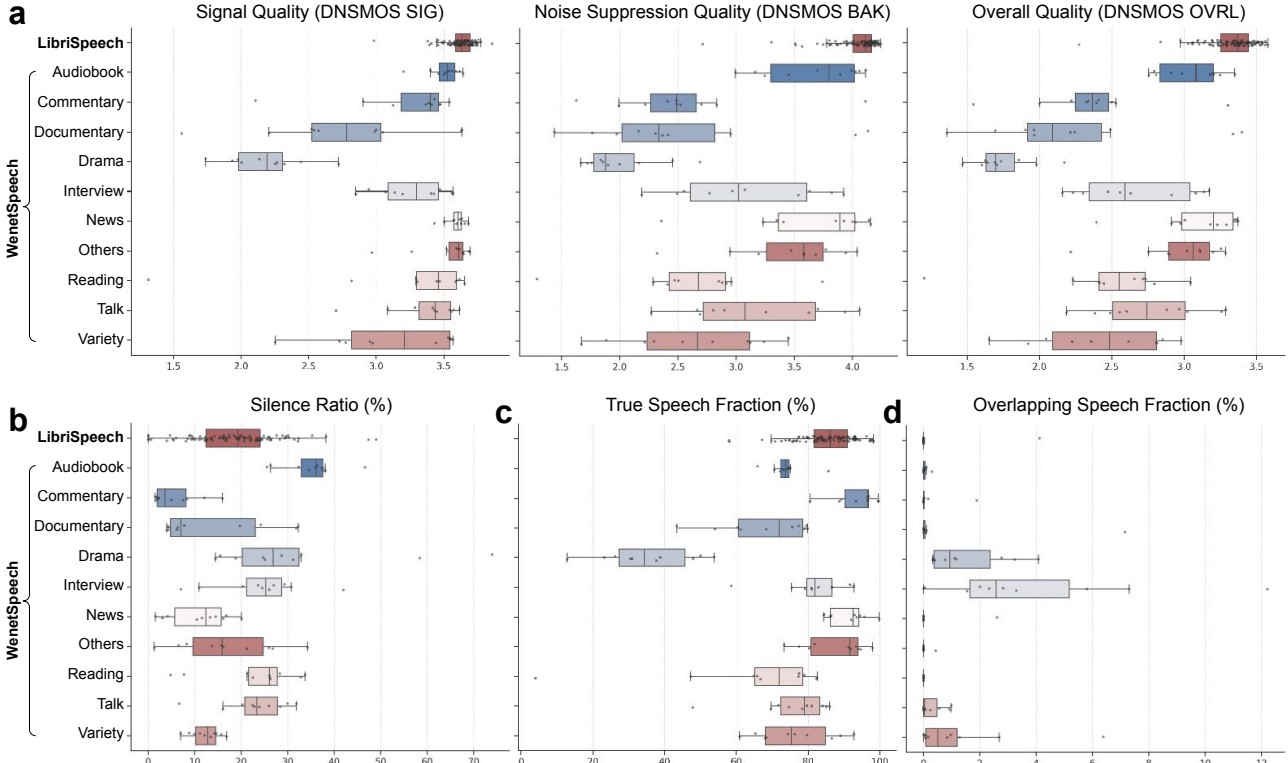

*Figure 10.* Quantification of audio quality and diversity between the LibriSpeech and WenetSpeech datasets. Notice the highly variable distribution of metrics among the 10 categories in WenetSpeech. We randomly chose 10 audio files from each of the 10 categories from the WenetSpeech dataset. We chose 50 audio files from LibriSpeech due to its shorter audio clip lengths. **a** Three metrics measure the signal quality, background noise suppression, and overall quality extracted by DNSMOS (Deep Noise Suppression Mean Opinion Score), a deep neural network trained on thousands of human ratings to "listen" to audio and output scores on a 1 to 5 scale (higher values are better). LibriSpeech (first row) exhibits higher quality than any category in WenetSpeech. In contrast, the Drama category, which constitutes 43% of all audio files in WenetSpeech, has the lowest quality. **b** The silence ratio, a low-level audio metric, represents the percentage of the audio file with less than 5% of the maximum energy (root mean square), calculated using Librosa, a Python library for audio analysis. Notice that the Commentary category has a silence period of less than 5%. **c** The true speech fraction measures the actual percentage of the audio that contains a human voice, completely ignoring loud background music or sound effects. We used a pretrained deep neural network for Voice Activity Detection (the Silero VAD model via PyTorch Hub) to compute this. Notice that the Drama category contains less than 40% speech time. **d** The overlapping speech fraction (extracted via speaker diarization) quantifies one of the defining features of diverse, real-world audio: the "cocktail party effect"—multiple speakers talking over one another. Domains featuring a single talker, like LibriSpeech and Audiobook, have nearly 0% overlapping speech. In contrast, conversational audio featuring multiple talkers, such as Talk, Drama, and especially Interview, has a higher percentage of time where two or more voices are active simultaneously. We used a pretrained model and pipeline called "pyannote.audio" to compute this fraction.

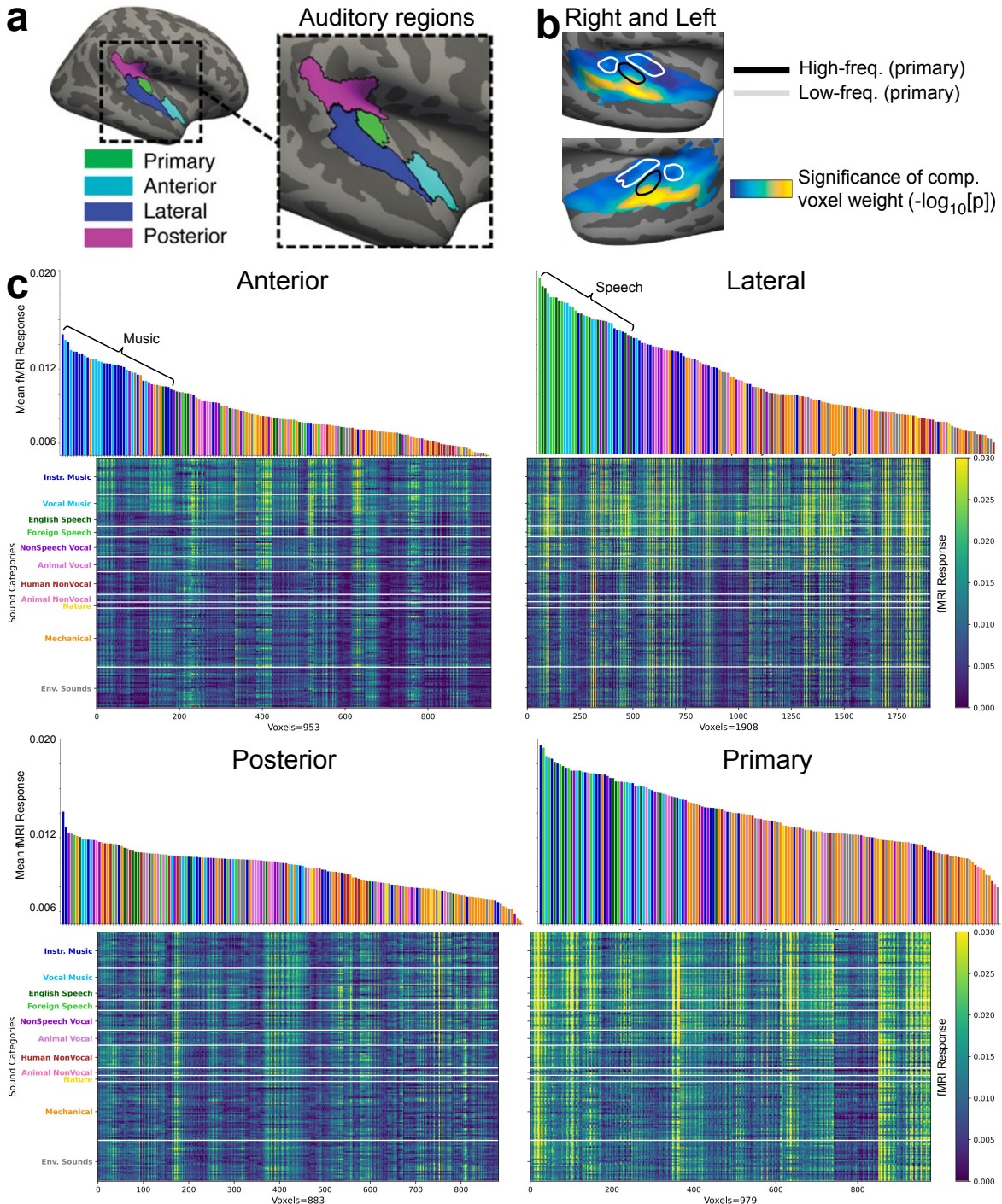

*Figure 11.* fMRI responses in the Lateral ROI show the strongest correlation with model activations. **a** Four anatomical regions of interest (ROIs) defined within the auditory cortex. **b** Spatial distribution of component voxel weights. Weights were averaged across 10 subjects aligned to a common anatomical template and converted to significance values via a permutation test. The color bar indicates logarithmically transformed $p$-values ($-\log_{10}[p]$). **c** The four panels display fMRI responses across four anatomically defined regions of interest (ROIs) within the human auditory cortex. In each panel, the top bar plots indicate the sorted, averaged fMRI responses across all voxels, with colors representing the 11 sound categories. The y-axis limits for the top bar plots and the color scale limits for the bottom heatmaps are fixed across all ROIs to facilitate direct comparison.

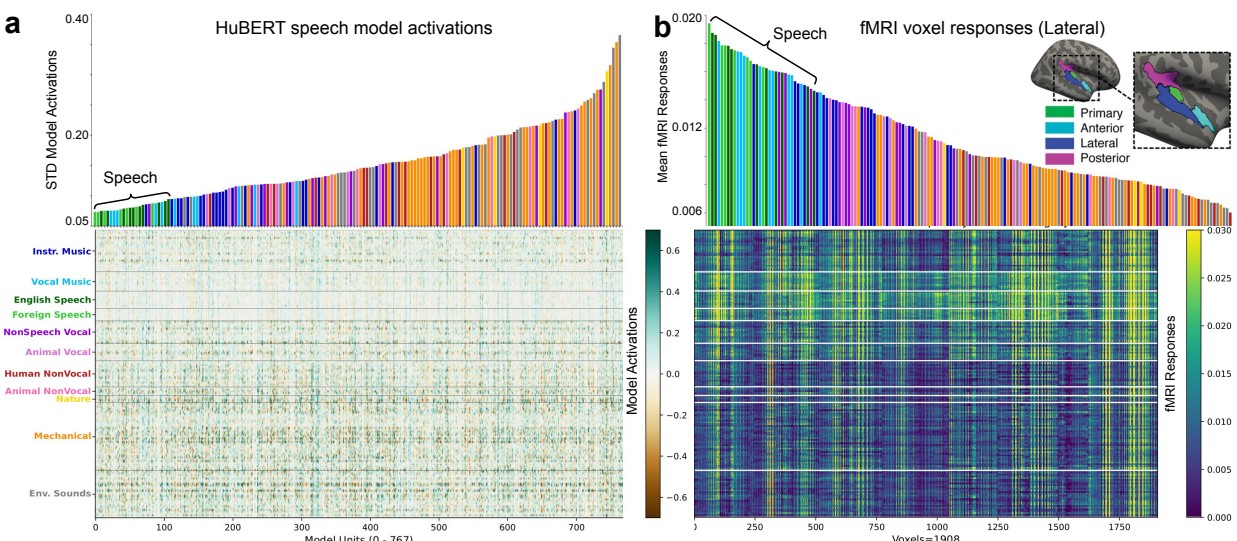

*Figure 12.* Tuning curve comparisons between HuBERTspeech model units and fMRI voxel responses. **a** Top: Sorted standard deviations across all 768 model units for each of the 165 sound stimuli. Bar colors indicate the 11 sound categories. Bottom: Model activations across all 768 units and 165 sounds. White indicates zero activation, while green and brown indicate maximum and minimum activations, respectively. **b** Top: Sorted mean fMRI responses across all 1,908 voxels in the Lateral area (blue patch, inset) of the human auditory cortex (right hemisphere). Bottom: fMRI responses across all voxels and sounds. Yellow and blue indicate maximum and minimum responses, respectively.

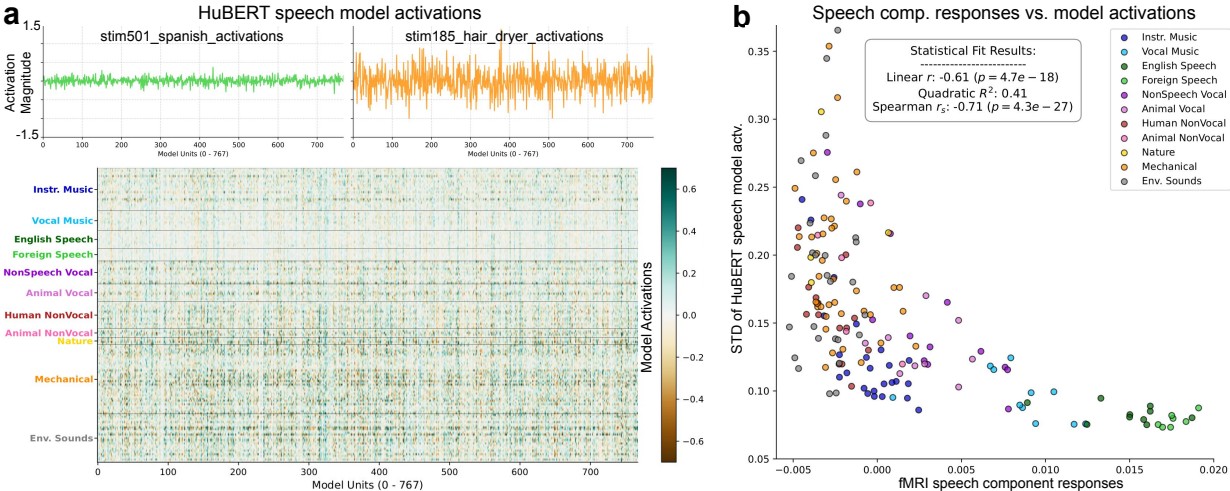

*Figure 13.* Tuning curve comparisons between HuBERTspeech model units and speech component responses. **a** Top: Activations of all model units in response to an example Foreign Speech sound (fresh green) and an example Mechanical sound (orange). Bottom: Activations of all model units across all 165 sounds (identical to Figure 12a, bottom). **b** Scatter plot comparing the standard deviation of model units (y-axis; identical values to 12b, top) against fMRI speech component responses (x-axis) for all 165 sounds. Note that Speech sounds (green) evoke stronger component responses and exhibit a smaller standard deviation in model activations.

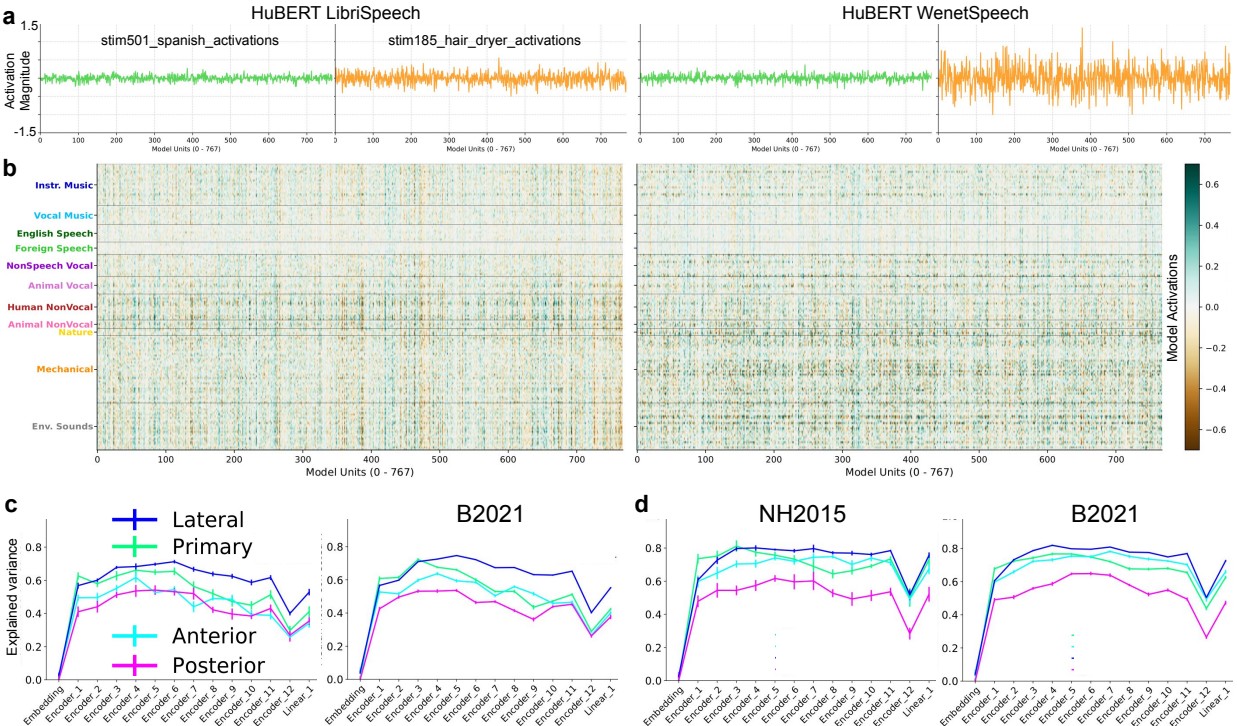

*Figure 14.* Differences in activation patterns between models explain variations in task performance **a** Model unit activations in response to two example sounds, formatted similarly to (left) or exactly as (right) Figure 13a, top. Note that the absolute magnitude and standard deviation (STD) differ between the two models: the HuBERTLS model, trained on clean speech (LibriSpeech), exhibits a smaller STD across its 768 units compared to the HuBERTspeech model, which was trained on real-world sounds (WenetSpeech). **b** Model unit activations to all 165 sounds, formatted similarly to (left) or exactly as (right) Figure 13a, bottom. Notice that the activation heatmaps for Music and Speech sounds are similar between the two models. The models primarily differ in their responses to 'Human NonVocal' and 'Nature/Mechanical/Environmental' sounds: the HuBERTLS model shows a larger STD for Human NonVocal sounds, while the HuBERTspeech model shows a larger STD for Nature/Mechanical/Environmental sounds. One explanation for this disparity is that the clean speech dataset primarily contains clean human non-vocal sounds, whereas the real-world speech dataset includes a diverse array of non-speech background sounds from surrounding environments. **c** Explained variance (y-axis) of the HuBERTLS model across all 14 layers in four regions of interest (ROIs) within the human auditory cortex, evaluated on the NH2015 (left) and B2021 (right) datasets. Both datasets yield consistent results: the model explains the largest variance in the Lateral ROI, an intermediate amount in the Primary and Anterior ROIs, and the smallest variance in the Posterior ROI. This aligns with the observations in Figure 3, where responses in the Lateral ROI are most highly correlated with the models, while the Posterior ROI shows the least correlation. Error bars represent within-participant standard error of the mean (SEM) and are smaller for the B2021 dataset due to its larger sample size (n = 20 vs. n = 8). **d** Same as (c), but for the HuBERTspeech model. Training on real-world sounds increases the explained variance across both datasets and all four auditory cortex ROIs, rather than being limited only to the Lateral ROI.

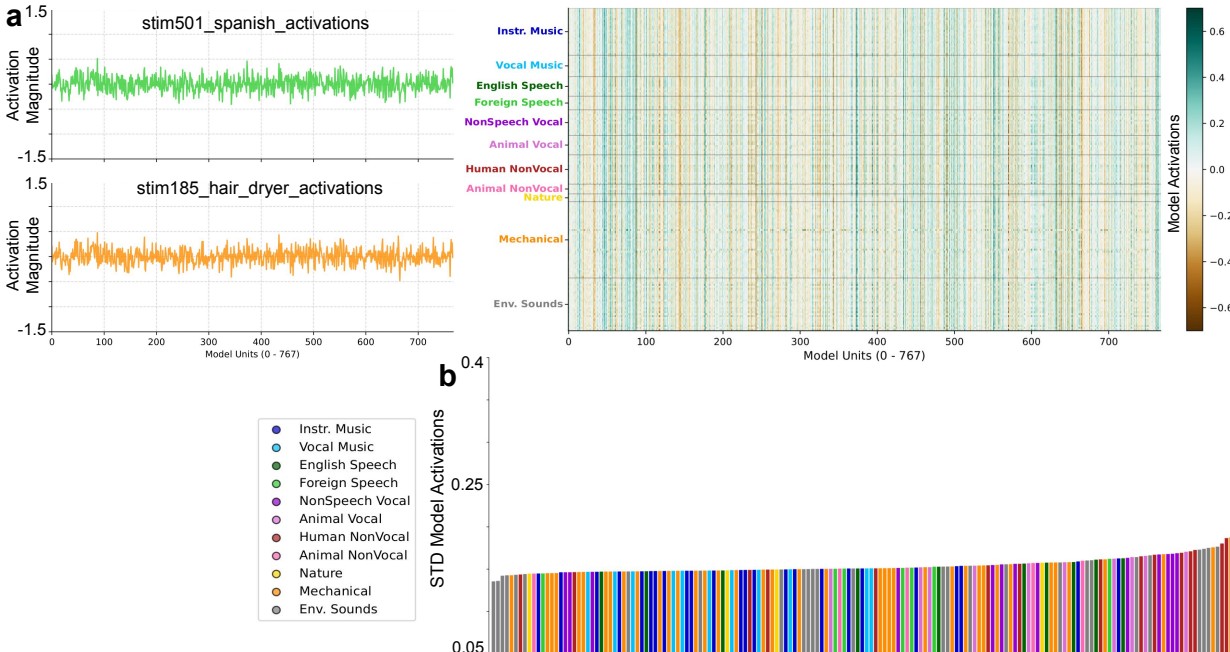

*Figure 15.* Unit activations in permuted models are unrelated to fMRI responses. **a** Formatted similarly to Figure 13a, but utilizing models with permuted network weights. Note that the activation amplitudes and standard deviations are similar both between examples (left) and across all 165 sounds (right). **b** Formatted similarly to Figure 12a. Note that the standard deviation of unit activations is small and uniformly distributed across all 165 sound stimuli for the permuted models.

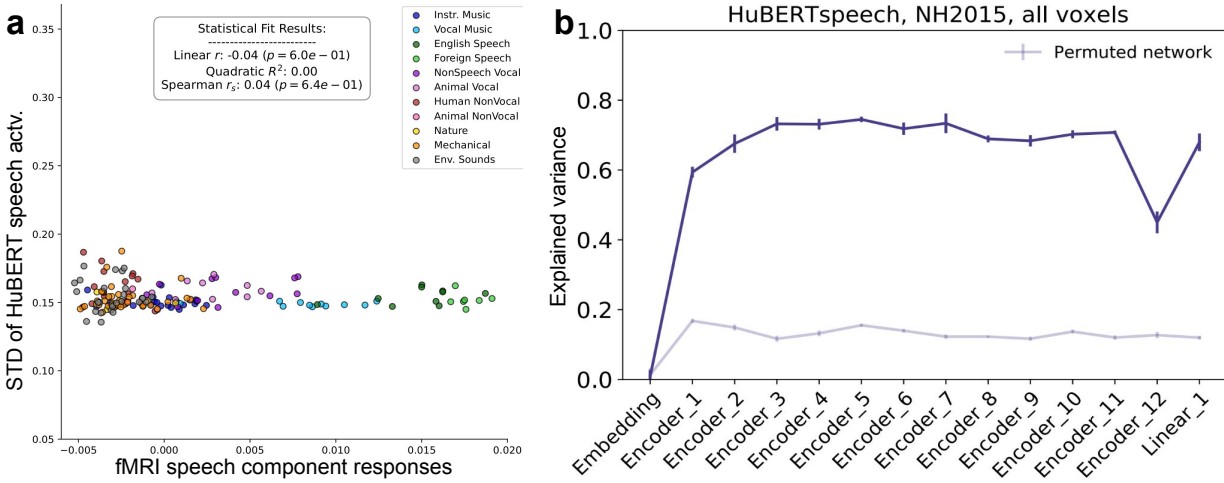

*Figure 16.* Unit activations in permuted models fail to explain fMRI component and voxel responses. **a** Formatted similarly to Figure 13b, with an identical x-axis but a flattened y-axis, indicating nonselective model activations in the permuted networks. **b** Formatted similarly to Figure 14c,d, but evaluated for voxels across the entire auditory cortex and including a weight-permuted network model for comparison. Error bars represent within-participant standard error of the mean (SEM).

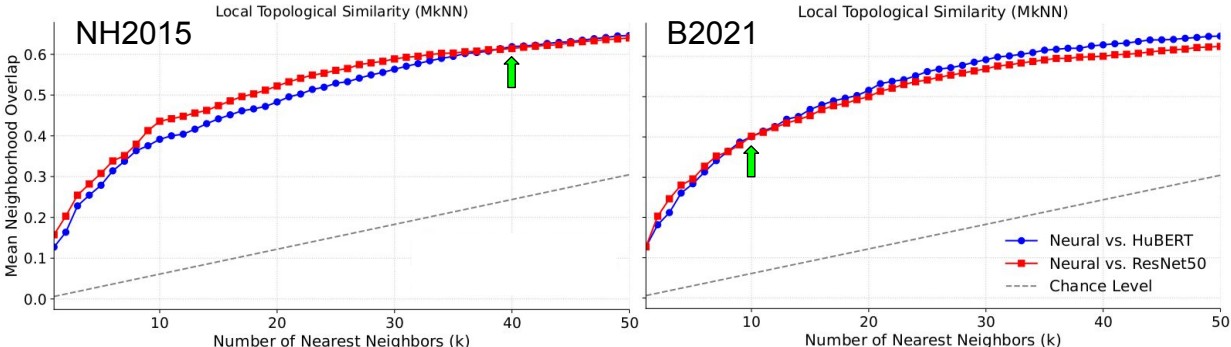

*Figure 17.* Quantitative comparison of RDMs from Figure 4 using Mutual k-Nearest Neighbors (MkNN). MkNN evaluates local topological geometry and neighborhood preservation between representational spaces. For each of the 165 sound stimuli, the overlap between its $k$ nearest neighbors in the neural RDM and its $k$ nearest neighbors in the model RDM was calculated. Mean neighborhood overlaps for both models significantly exceed chance levels (gray dashed line) across all evaluated neighborhood sizes. At small neighborhood sizes (green arrow; $k < 40$ for the NH2015 dataset; $k < 10$ for B2021), ResNet50multitask (red) shares a higher proportion of immediate neighbors with the neural data, indicating stronger alignment with fine-grained, low-level acoustic clustering. Conversely, as the neighborhood expands ($k > 40$ for NH2015; $k > 10$ for B2021), the curves intersect and HuBERTspeech (blue) demonstrates greater overlap. This suggests that HuBERTspeech better captures the broader, higher-level categorical structures of the neural representations.

# C. Methods

### C.1. HuBERTspeech and Wav2Vec2speech pretrained models

To ensure the model learned representations robust to the statistics of natural auditory environments, we utilized the WenetSpeech corpus (Zhang et al., 2022). Unlike standard curated datasets such as LibriSpeech, which are restricted to clean, read speech (audiobooks) in sterile acoustic conditions, WenetSpeech is harvested from unconstrained "in-the-wild" internet sources (YouTube and podcasts). The dataset comprises 22,400+ hours of audio, including a core subset of 10,000+ hours of high-confidence labeled speech (confidence $\geq 95\%$).

Crucially, the data spans 10 diverse domains (Fig. 9, 10), including Drama, Variety Shows, Interviews, News, Sports, and Commentary—thereby introducing substantial variance in both speaking style and acoustic quality. This diversity exposes the model to spontaneous, non-scripted speech, emotional prosody, overlapping talkers, and significant background noise (e.g., music, environmental sounds). We posit that this exposure to "suboptimal" and highly variable acoustic statistics is not a detriment but a key computational advantage: by adapting to the noisy, high-contrast density of real-world sensory inputs, the model is forced to develop invariant and robust feature representations, mirroring the biological imperative where the brain optimizes itself against the unconstrained statistics of the natural world.

Pretrained checkpoints for models trained on the WenetSpeech corpus were obtained from the TencentGameMate repository on HuggingFace (Guo & Liu, 2022) (`https://huggingface.co/TencentGameMate`). The repository provides four unsupervised architectures: HuBERT and Wav2Vec2, in both Base and Large configurations. For this study, we utilized the Base versions of each architecture. To distinguish these models from their standard counterparts, we refer to them herein as HuBERTspeech and Wav2Vec2speech, respectively. Details on the models' original training process can be found at: https://github.com/TencentGameMate/chinese_speech_pretrain. As per the developers, the Base model was trained using 8 A100 GPUs, and the Large model used 16 A100 GPUs.

### C.2. HuBERTLS/Wav2Vec2LS/HuBERTcore pretrained models

Wav2Vec2 pretrained on LibriSpeech (Wav2Vec2LS) model was downloaded from the Facebook repository on HuggingFace (`facebook/wav2vec2-base`). HuBERTLS was also downloaded from the Facebook repository on HuggingFace (`facebook/hubert-base`).

HuBERTcore (referred to as AVES-core in the official documentation)(Hagiwara, 2023) is a self-supervised, transformer-based audio representation model designed to encode diverse soundscapes. It leverages the HuBERT base architecture, which consists of a CNN feature extractor followed by a 12-layer Transformer encoder with an embedding dimension of 768 and approximately 95 million parameters. This architecture operates on 16 kHz raw audio waveforms and is trained using a masked prediction objective, where the model learns to predict discrete acoustic units (pseudo-labels generated via K-means clustering) from masked parts of the input.

The pretrained dataset for HuBERTcore is a curated collection of unannotated audio designed to establish a fundamental understanding of general acoustic events. Specifically, it is pretrained on FSD50k (Freesound Dataset 50k)(Fonseca et al., 2021) combined with a core subset of AudioSet (Gemmeke et al., 2017). This combination results in a total pretraining duration of approximately 153 hours. Pretrained model is available at: `https://github.com/earthspecies/aves`.

### C.3. AST/VGGish/S2T/Wav2Vec2FT pretrained models

The Audio Spectrogram Transformer (AST)(Gong et al., 2021) is a purely attention-based model adapted from the Vision Transformer architecture for audio classification tasks. Instead of raw waveforms, AST takes a log-mel spectrogram as input, which is split into a sequence of patches that are linearly projected into 1D embeddings. These embeddings, combined with a positional encoding, are processed by a series of standard transformer encoder blocks. We utilized a model pretrained on the AudioSet dataset, extracting representations from the initial embedding layer, the outputs of the 12 transformer encoder blocks, and the final linear projection layer.

VGGish (Hershey et al., 2017) is a CNN inspired by the VGG architecture originally developed for image recognition. It is trained to classify audio events using the large-scale AudioSet dataset, which consists of over 2 million human-labeled YouTube video clips covering 527 audio classes. The model accepts log-mel spectrograms as input and processes them through a series of convolutional and max-pooling layers. For our analysis, we extracted unit activations from the model's

layers to capture the hierarchical features learned during the audio event classification task.

S2T (Speech-to-Text)(Wang et al., 2020) is an encoder-decoder model designed for end-to-end speech recognition. The architecture accepts log-mel spectrograms as input, which are first processed by two convolutional layers before being passed through a stack of 12 transformer encoder blocks. The model was trained using a cross-entropy loss function to generate a transcript from the input audio, utilizing a vocabulary of unigram tokens. For our analysis, we focused on the encoder portion of the model, extracting activations from the initial embedding layer and the subsequent 12 transformer encoder blocks.

Wav2Vec2FT (referring to the fine-tuned Wav2Vec2 model)(Baevski et al., 2020) is a transformer-based architecture designed for automatic speech recognition that operates directly on raw audio waveforms. The model consists of a multi-layer convolutional feature encoder that processes the waveform, followed by a stack of transformer encoder blocks that model temporal dependencies. While the core model is pre-trained via a self-supervised contrastive task to distinguish true speech representations from latent distractors, the fine-tuned version (Wav2Vec2FT) includes an additional linear projection layer and is optimized using a Connectionist Temporal Classification (CTC) loss on labeled speech data (e.g., LibriSpeech). We extracted representations from the initial embeddings, the outputs of the 12 transformer blocks, and the final character class logits.

### C.4. CochCNN9/ResNet50 Word/Speaker/AudioSet/MultiTask/Genre pretrained models

We evaluated two distinct "in-house" model architectures, CochCNN9 and CochResNet50, to examine the effects of architecture and training task on brain predictivity. Both architectures utilize a biologically inspired "cochleagram" input stage, which consists of a time-frequency representation generated by passing audio through a bank of 211 filters spaced according to the Equivalent Rectangular Bandwidth (ERB) scale, followed by a compressive nonlinearity to mimic the human ear. CochCNN9 is a deep convolutional neural network consisting of five convolutional layers followed by fully connected layers, replicating the architecture used in previous auditory neuroscience research (Kell et al., 2018). CochResNet50 adapts the standard ResNet50 computer vision architecture to accept cochleagram inputs, allowing for deeper feature extraction through residual connections.

To investigate the role of task constraints, these architectures were trained on a Word-Speaker-Noise dataset constructed to support three distinct classification tasks using the same audio exemplars. The dataset combines speech clips from the Wall Street Journal and Spoken Wikipedia Corpora with superimposed environmental sounds from AudioSet. The Word models were optimized to recognize the word at the center of the clip, the Speaker models were trained to identify the talker's identity, and the AudioSet (or Audio Event) models were trained to classify the background environmental sounds. This design ensured that differences in model representations were driven by the task objective rather than differences in the training stimuli.

We also analyzed MultiTask variants of both architectures, which were jointly optimized to perform all three tasks (Word, Speaker, and Audio Event recognition) simultaneously. These models shared the initial feature extraction layers but branched into separate readout heads for each task. This approach allowed us to determine whether a representation optimized for multiple concurrent auditory goals provides a better account of cortical responses compared to models specialized for a single domain.

Finally, we included models trained on a Musical Genre classification task to contrast speech- and event-based training with music processing. These models were trained on a separate dataset of musical clips covering 41 distinct genres, as originally compiled by (Kell et al., 2018). By evaluating the CochCNN9 and CochResNet50 architectures across this diverse set of tasks—ranging from speech and environmental sound recognition to music classification—we sought to dissociate the contributions of model architecture and training domain to the prediction of human auditory cortical activity.

### C.5. camSAY pretrained models

We utilized the SAYCam (Sam, Alice, and Yale/Asa) dataset, a large-scale, longitudinal, audiovisual corpus recorded from the egocentric perspective of three infants (Sullivan et al., 2021). The dataset captures the naturalistic visual and auditory environment of three developing children (labeled S, A, and Y) typically developing in English-speaking households in Australia and the United States. Recordings were collected using head-mounted Veho MUVI Pro cameras equipped with fisheye lenses (109° horizontal × 70° vertical field of view), yielding video at 640×480 resolution. The data spans ages from approximately 6 to 32 months, collected at a frequency of roughly two hours per week. In total, the dataset comprises

over 415 hours of unscripted footage documenting diverse daily activities, including playing, eating, and transit, in various settings such as homes, cars, and outdoor environments.

For visual feature extraction, we employed the pretrained "Baby Vision" model (Orhan et al., 2020), specifically the updated ResNeXt-50 (32x4d) architecture released in the associated repository. Unlike standard ImageNet-pretrained models, this network was trained entirely via self-supervision on the raw video streams from the SAYCam dataset. The model utilizes a Temporal Classification (TC) objective, where the continuous video stream is discretized into temporal windows, and the network is trained to classify individual video frames into their corresponding temporal segments. This objective encourages the model to learn robust representations that are invariant to short-term transforms but discriminative of distinct temporal events. The model (TC-SAY-resnext) was trained on the combined data from all three children for 16 epochs with a batch size of 256, utilizing a data augmentation pipeline similar to SimCLR that includes random resized crops, color jittering, Gaussian blur, and random horizontal flips. The four pretrained models are availble from: https://github.com/eminorhan/baby-vision.

### C.6. Stimuli related to Fig. 7 and 8

Coggan & Tong (2023) measured high-field 7T fMRI responses in 10 human subjects. The stimuli were derived from a study by (Bao et al., 2020) and were specifically designed to span a feature space defined by deep neural network representations. The primary 'spikiness-animacy' set consisted of 80 object images selected based on the first two principal components of AlexNet activations (fc6), which corresponded to spikiness and animacy, respectively. These images were divided into four distinct conditions (20 images each): spiky-animate, spiky-inanimate, stubby-animate, and stubby-inanimate. To facilitate the localization of category-selective regions, the dataset also included a 'classic category' set consisting of 80 images of faces, bodies, houses, and common objects. All stimuli were converted to grayscale, presented on a white background, and subtended a visual angle of approximately 9°.

For the Bracci et al. (2019), the visual stimuli consisted of a carefully curated set of 27 color images designed to dissociate object appearance (visual features) from object category (animacy). These images were organized into nine distinct "triads." Each triad contained three specific types of items: (1) a living animal (e.g., a cow), (2) a typical inanimate object (e.g., a mug), and (3) a "lookalike" object (e.g., a cow-shaped mug). The lookalike condition was the critical manipulation: these items were inanimate artifacts that shared the functional identity and size of the regular objects (both are mugs) but shared the visual appearance (shape, texture, and features) of the animals.

### C.7. Extraction of model activations

Activations for the five newly added models were extracted following the same protocol as the Wav2Vec2 model (renamed here as Wav2Vec2FT) originally utilized by Tuckute et al. (2023). We extracted deep neural representations from 165 natural sounds using the hubert-base or wav2vec2-base model. Pre-processing was handled via the Wav2Vec2FeatureExtractor, which standardized all audio inputs by resampling them to 16 kHz, downmixing to mono, and normalizing waveforms to zero mean and unit variance. The model architecture consists of a convolutional feature encoder (7 layers) followed by a Transformer context encoder (12 layers). We targeted 14 distinct layers to capture the hierarchical processing of auditory information, ranging from early sensory features to high-level semantic representations.

The first extraction point was the initial layer of the feature extractor, a 1-dimensional convolution ($Conv1d$) with a kernel size of 10, a stride of 5, and 512 output channels (Conv1d(1, 512)). Subsequent activations were extracted from the Transformer encoder, which consists of 12 identical blocks. Within each block, we tapped the output of the feed-forward network's projection layer. These linear layers transform the internal representation from an intermediate dimension of 3072 back to the model's hidden dimension of 768 ($Linear(3072, 768)$). We sampled this projection layer from every Transformer block (indices 0 through 11) alongside the model's final hidden state.

To generate a fixed-length feature vector for each sound, we applied global average pooling across the temporal dimension for all extracted layers. This reduced the time-varying activation maps to static vectors representing the aggregate spectral and semantic content of the stimulus. To control for architecture-driven correlations, we implemented a "random network" baseline (randnetw). In this condition, all model weights—including convolutional kernels and linear projections—were re-initialized to random values prior to feature extraction, preserving the architecture while removing learned statistical regularities.

## C.8. Voxel response modeling or encoding model

General approach. To evaluate how well candidate models account for human auditory cortical activity, we performed a voxelwise encoding analysis. We treated each layer of a deep neural network (DNN) as a potential hypothesis for the computational implementation of auditory processing. Given the slow temporal dynamics of the hemodynamic blood-oxygen-level-dependent (BOLD) signal measured by fMRI, we focused on predicting the time-averaged response of each voxel to each sound stimulus, rather than tracking fine-grained temporal dynamics. Correspondingly, we averaged the activations of the model units over the temporal dimension to generate a single feature vector per sound for each model layer. For architectures containing Rectified Linear Units (ReLU) or Tanh non-linearities, we extracted activations after the non-linearity to capture the rectified output. For Transformer-based architectures (e.g., AST, Wav2Vec2) which lack these specific stage boundaries, we extracted the real-valued unit activations directly.

Regularized linear regression and cross-validation. We modeled the response of each voxel as a linear combination of the time-averaged unit activations from a specific model stage. Because the number of model units (regressors) typically exceeded the number of audio stimuli (observations), we employed L2-regularized linear regression (Ridge regression) to fit the mapping from model features to brain responses. We used a cross-validation procedure to evaluate the models on unseen data. The fMRI data were split into training and testing sets; within the training set, we performed an inner loop of cross-validation to optimize the regularization parameter ($\lambda$) for each voxel. The weights derived from the training set were then applied to the model features of the held-out test sounds to generate predicted voxel responses.

Summary of evaluation. The predictive performance of each model stage was quantified by comparing the predicted voxel responses to the actual measured fMRI responses in the held-out test set. We calculated the Pearson correlation coefficient between the predicted and observed time courses for each voxel. To account for measurement noise in the neural data, we computed the squared correlation and normalized it by the voxel's noise ceiling—an estimate of the maximum explainable variance given the reliability of the fMRI signal. This yields a measure of the noise-corrected explained variance, indicating how well the features of a given model stage capture the systematic variance in the auditory cortical responses.

## C.9. Representational similarity analysis

To ensure the robustness of our results across different evaluation metrics, we complemented the regression-based encoding analysis with Representational Similarity Analysis (RSA) (Kriegeskorte et al., 2008). We utilized the same set of model stages and time-averaged feature representations as described in the voxelwise modeling section. For each model stage, we constructed a model Representational Dissimilarity Matrix (RDM) by calculating the pairwise dissimilarity between the unit activation vectors elicited by each pair of sounds. Dissimilarity was defined as $1 - r$, where $r$ is the Pearson correlation coefficient. Analogously, we constructed an fMRI RDM for each participant by computing the dissimilarity ($1 - r$) between the voxel response patterns for each stimulus pair. Prior to RDM construction, both the model unit activations and the fMRI voxel responses were z-scored. We quantified the similarity between the model and the human auditory cortex by computing the Spearman rank-order correlation coefficient between the model RDM and the fMRI RDM.

For the aggregate comparisons (Fig. 3), we sought to determine how well the best-performing stage of each model captured the representational structure of the auditory cortex. To avoid overfitting the stage selection, we employed a cross-validation procedure. We generated 10 independent random splits of the 165 sound stimuli, dividing them into a training set ($n = 83$) and a testing set ($n = 82$). For each split, we first calculated the RDMs for the training sounds and identified the specific model stage that yielded the highest Spearman correlation ($\rho$) with the participant's fMRI RDM. We then validated this selection on the held-out test set by computing the Spearman $\rho$ between the RDM of the selected model stage and the fMRI RDM derived from the test sounds. The final performance metric for each model was calculated as the median Spearman $\rho$ across the 10 cross-validation folds. This process was repeated for every candidate model across all participants in both datasets (NH2015: $n = 8$; B2021: $n = 20$), and the results were averaged across participants. For comparison, we applied this identical evaluation protocol to the SpectroTemporal baseline model and to permuted control versions of each neural network.

