# OpenReview forum: "Real-World Unsupervised Models Generalize to Predict Brain Responses to Out-of-Distribution Stimuli"
_ICML.cc/2026/Conference — ICML 2026 spotlight_

### Official Review · Reviewer_ps8P · 2026-03-10

**Soundness:** 2
**Presentation:** 2
**Significance:** 3
**Originality:** 3
**Overall Recommendation:** 4
**Confidence:** 2

**Summary:**

The authors study the interplay of parameters like architecture, objective (supervised or unsupervised), data size and data variability to predict and replicate brain responses (using fMRI) in the auditory and visual cortex for audio and visual stimuli respectively.​ They found that unsupervised models trained on a varied and large dataset give SOTA performance for both audio and video ​

**Compliance With Llm Reviewing Policy:**

Affirmed.

**Final Justification:**

After reading the rebuttal reply and discussions with other reviewers, I have decided to increase my score to weak accept. The authors provide a strong rebuttal addressing major concerns of all reviewers. My change of score is based assuming that the authors will remove some inaccurate claims made in the paper and support the remaining claims with additional evidence from the rebuttal. Also, clearly discussing connections to NeuroAI and biological plausibility would greatly improve the manuscript.

**Key Questions For Authors:**

- In visual data analysis using the dataset from Bracci et al. (2019), the unsupervised models trained on SAYCam dataset are compared with ImageNet-pretrained models. Are the latter models unsupervised ? ​

- Similarly, the details of the baselines used from Brain-Score benchmarks is not clear. What is the training objective (supervised or unsupervised) , data size and data variability of these models? (Also, the x-axis text (model names) in the corresponding figure is very small). The SOTA models in the Brain score benchmark paper achieve a score of 0.549 (lowest being 0.454) which is higher than the baseline models and in house models presented in the study, is there a specific reason for not including these models in the benchmarking ? ​

- “This finding plausibly explains why models like HuBERTspeech and Wav2Vec2speech— despite being pretrained on WenetSpeech, a Mandarin corpus—can accurately predict brain responses in native English speakers” – This is the main finding of the paper. But referring to Figure 5 and prior similar analysis with supervised models (Tuckute et al. (2023)) , It is not clear to me how this is different from previous supervised models (specifically, CochResNet50-MultiTask) ?

**Limitations:**

Limitations of this study have not been addressed.

**Strengths And Weaknesses:**

Strengths:

- The authors have used previous findings about noisy data aiding models to better mimic brain responses, to construct the hypothesis that unsupervised models trained on noisy data can mimic brain responses better than supervised models. Consequently, they show that such models can achieve SOTA in predicting and representing brain responses. ​

- The study uses well established metrics and compares the results using two architectures (HuBERT and Wav2Vec2) trained with an unsupervised objective on datasets of different sizes and variability and thus adds to the existing benchmarks (of supervised models) for audio cortex.​

- The HuBERT speech model seems to capture the shared “vocal” feature in the brain better that the SOTA supervised models. (Quantitative analysis in addition to visual inspection can strengthen this claim further).

Weaknesses:

- While unsupervised models trained on large variable data is shown to perform better than SOTA, some claims are neither backed with a systematic comparison nor a quantitative analysis.​

   - For example, “data distribution (real-world vs. manually curated) contributes more significantly to brain prediction than dataset size”. Here, HuBERTCore (trained with small and real-world data) is compared with HuBERTLS (trained with large but manually curated data) only in one of the two datasets (NH2015) without any quantitative measure of how much better HuBERTCore is compared to HuBERTLS. Additionally, the second dataset (B2021) shows evidence that contrasts this claim (HuBERTLS is actually better than HuBERCore). ​

  - While comparing the topography of brain responses, the authors claim that CochResNet50-MultiTask, the SOTA supervised model deviates significantly from the neural data – Was there a statistical test performed for this conclusion ? Also, this model ranks third and second in terms of Spearman correlation (Figure 3 and Figure 2 respectively), which indicates the representational similarity. This signals a contradiction to the claim that it deviates from the neural data. ​

   - “HuBERTspeech best predicts the speech-selective neural component” – Questions like why the speech component was picked from among the 6 components is unanswered ​

- Is not a self-sustained study - The definition of key metrics like RSA and methods of calculating predictions, conclusions from Figure 5 is not very clear. Some figures in the Appendix (Figure 9, 10, 11) must be a part of the main text. ​

- The analysis with visual data needs to be more elaborate. For example, explain the baseline models used for benchmarking.

---

> ### Author Rebuttal · Authors · 2026-03-31
>
> We sincerely thank the reviewer for their constructive feedback!\
> Below is an anonymous link to a PDF containing Rebuttal figures and captions:\
> https://anonymous.4open.science/r/21034/Rebuttal.pdf
>
> **W1-1**\
> We sincerely apologize for making these claims without providing a sufficiently rigorous and quantitative data analysis.
>
> Rebuttal Table 1 provides the statistical comparison between the HuBERTCore and HuBERTLS models. The reviewer is correct that for the B2021 dataset, using brain prediction accuracy ($r^2$) as the metric, HuBERTLS outperforms HuBERTcore. However, the opposite trend was found in the remaining three conditions. For example, in the B2021 dataset with the RSA metric, HuBERTcore outperforms HuBERTLS by 15.5%. We will include this table in the Appendix and only report the values instead of making inaccurate claims.
>
> **W1-2**\
> We apologize for this inaccurate claim and have removed the sentence, as it lacked quantitative support. Instead, we now compare the RDMs in Fig. 4 (neural and HuBERTspeech) and Fig. 11 (neural and CochResNet50-MultiTask) using two metrics suggested by Reviewer j29Q (please refer to our response to W3).
>
> **W1-3**\
> Please refer to our responses to W2-2 and Q3.
>
> **W2-1 Definition of metrics**\
> We will add the definition of RSA "Specifically, we employed cross-validated, L2-regularized linear regression to predict time-averaged voxel responses from the time-averaged unit activations of each model layer." and methods of calculating predictions "Representational Dissimilarity Matrices (RDMs) using the pairwise Pearson correlation distance (1 - $r$) between stimulus-evoked response patterns, and quantified model-brain alignment via the Spearman rank correlation between the model and fMRI RDMs."
>
> **W2-2 Conclusions from Fig. 5**\
> Fig. 5 examines the prediction of speech components and complements previous Figs. 2-4, which focus on metrics and topology of all sounds. It provides further evidence of the generalization of the real-world pretrained model to OOD English speech. Rebuttal Fig. 2  shows that model unit activations are significantly correlated with speech component responses. Since the weight of the speech component was highest in the Lateral auditory cortex (Norman-Haignere et al., 2015), the tuning curves of model units match the fMRI voxel responses mainly in the Lateral area (Rebuttal Figs. 1 and 3). Together, Fig. 5 and Rebuttal Figs. 1-3 demonstrate and further explain the HuBERTspeech model’s generalization to OOD stimuli.
>
> **W2-3 Appendix Figs.**\
> We will merge Figs. 9-11 with the current Figs. 2-4, exactly as shown in Rebuttal Figs. 9-11.
>
> **W3**\
> Please refer to our response to Q1 and Q2.
>
> **Q1**\
> We thank the reviewer for this question; please refer to our response to Q2 from Reviewer u4ti.
>
> **Q2-1 Details of baselines**\
> Providing an exhaustive list of every Brain-Score baseline's training details is prohibitive. The vast majority of these models, including 20 of the top 22 in Fig. 7 and all models highlighted in Fig. 8, utilize supervised training on the curated ImageNet-1K dataset (1.2M images). The two exceptions (vit_large and convnext_large) use SSL on massive, highly variable image-text datasets (400M and 2B pairs, respectively). We will add these representative details to Appendix B.5.
>
> **Q2-2 X-axis text**\
> We will move the inset panel of Fig. 7 to the Appendix and enlarge the text in Fig. 8c.
>
> **Q2-3 SOTA models**\
> We thank the reviewer for raising this question. To clarify the metric in question, the SOTA model (DenseNet-169) from the Brain-Score paper (Schrimpf et al., 2018; Table 1) has a Brain-Score of 0.549, which is the average of its neural predictivity of V4 (0.663), IT (0.606), and behavioral predictivity (0.378). The IT score is calculated as the average across seven datasets, two of which are utilized in Figs. 7 and 8.
>
> To mechanistically explain model behavior, we require tractable comparisons. The auditory benchmark's constrained scale (19 models, Table 1) allows us to cleanly isolate architecture, objective, and dataset impacts. In contrast, the visual Brain-Score’s massive heterogeneity (400+ models) prevents effective variable control. Consequently, we avoided these confounding variables by restricting our Figure 8 visual baselines to highly representative, well-understood models: supervised ResNeXt and the SSL DINO suggested by the reviewer.
>
> **Q3**\
> We apologize for the lack of clarity. The key distinction is domain alignment. The CochResNet50-MultiTask model was trained on English, matching the native English fMRI subjects (ID). Conversely, our HuBERTspeech was pretrained exclusively on Mandarin, making the English stimuli profoundly OOD. Despite this OOD mismatch, our Mandarin-trained model (our Fig. 5, $R = 0.88$) actually outperformed the domain-matched English supervised model (Tuckute et al., 2023, Fig. 4A, $R = 0.87$).
>
> **Limitations**\
> We will add a "Limitations" section as noted in our response to Reviewer iMjN.

---

> > ### Author Rebuttal · Reviewer_ps8P · 2026-04-02
> >
> > I appreciate the authors responses to my concerns and for providing the updated figures. However, I still have some concerns.
> >
> > First, in the rebuttal figures 9 and 10, the original figures from the paper are combined but the figures seem to share the x-axis. Looking at the original figures from the manuscript it seems incorrect for figure 2, 9 and figure 3,10 to share the x-axis. I suppose each model is represented by a different color in which case the x-axis should be removed. Could you please clarify this ?
> >
> > Second, following the revisions to claims W1-1 and W1-2, it would be helpful if the authors could explicitly summarize the updated findings. In particular, what are the key conclusions that can now be drawn from the revised quantitative analysis? A clear statement of the revised insights would improve the overall clarity and impact of the manuscript.

---

> > > ### Author Response · Authors · 2026-04-02
> > >
> > > We appreciate the reviewer’s attention to detail. Because the bars in each subplot were sorted independently by their performance values, the specific order of the models varies across the subplots, making a shared x-axis misleading.
> > > Following your suggestion, we have updated Rebuttal Figs. 9 (https://anonymous.4open.science/r/21034/Reviewer_ps8P_Rebuttal_Fig9.pdf) and 10 (https://anonymous.4open.science/r/21034/Reviewer_ps8P_Rebuttal_Fig10.pdf) to remove the shared x-axes. Instead, the 19 models (listed in the same order as Table 1) are now consistently identified across all subplots by their assigned colors, as shown in the legend at the bottom.
> > >
> > > We agree that explicitly summarizing the updated findings will improve the overall clarity and impact of the manuscript. The shift from our original qualitative claims (W1-1 and W1-2) to our revised quantitative analyses (RSA, MkNN, and CKA) yielded refined conclusions regarding how data distribution and learning objectives shape model-brain alignment. We will add the following paragraph to the Discussion:
> > >
> > > *Our quantitative analyses reveal two key insights into model-brain alignment. First, regarding training data, we observe a functional trade-off: while massive, curated datasets (HuBERTLS) can sometimes yield higher raw voxel-wise predictivity, pretraining on a much smaller, uncurated, real-world dataset (HuBERTcore) consistently yields superior global representational alignment (measured by RSA). This suggests that naturalistic data distributions serve as a highly efficient inductive bias for capturing the brain's global topological geometry. Second, regarding learning objectives, our analyses using MkNN and CKA indicate that supervised and self-supervised models optimize for different levels of the auditory processing hierarchy. Supervised models (e.g., CochResNet50-MultiTask) excel at capturing fine-grained, low-level acoustics, whereas self-supervised models (e.g., HuBERTspeech) better capture the global geometry and broad categorical structure of the human auditory cortex.*
> > >
> > > We appreciate your continued engagement. Your feedback has greatly enhanced the rigor, clarity, and impact of our manuscript. As this is the final interaction phase, we hope these updated figures and the summary of insights fully resolve your remaining concerns. We look forward to your updated assessment of our work in light of these revisions.

---

### Official Review · Reviewer_u4ti · 2026-03-11

**Soundness:** 4
**Presentation:** 3
**Significance:** 3
**Originality:** 2
**Overall Recommendation:** 5
**Confidence:** 3

**Summary:**

This paper extends a study from 2023 towards understanding the similarity of latent representations between DNNs and the human brain (fMRI) in response to auditory and visual stimuli. For audio, It presents an analysis over a set of pretrained models . For video, the author compare pretrained baby-vision models with ImageNet pretrained models. In both cases, the conclusion seems to be that unsupervised pretrained models produce representations better aligned with brain activity. Two metrics were used: model-based-prediction, and RDA.

Overall, the analysis presented is quite convincing towards the authors’ hypothesis. I am open to increasing my score to a 5 once my questions and concerns are adequately addressed.

**Compliance With Llm Reviewing Policy:**

Affirmed.

**Final Justification:**

All my concerns are addressed. The authors promise to move some things from the appendix into the main text so it looks more like a normal ML paper. Plus, they have provided a new comparison, which they will add in the paper.

**Key Questions For Authors:**

1. Why does CochCNN9/ResNet50 have such strong performance, even though it is a supervised model?

2. In figure 8, how were the `resnext*_imagenet` models trained? I suspect these were supervised models. If this is true, then I don’t believe it to be a fair comparison. A model trained with SSL (e.g. DINO) would be a better point of comparison with the SSL-trained baby-vision models.

3. The following statement:


> Furthermore, HuBERT-Speech achieved higher variance explained (r2) than the previous best supervised model (CochResNet50-MultiTask) in 8 out of 10 participants.

seems inconsistent with the earlier description of the NH2015 dataset containing 8 participants.

**Limitations:**

yes

**Strengths And Weaknesses:**

#### Strengths

1. The introduction is written quite well. It makes a pretty convincing argument for why models learned with unsupervised methods on naturalistic long-tailed datasets would produce representations similar to the human brain.

2. A good amount of models are included in the audio data case.


#### Weaknesses

1. The paper seems to be formatted for a journal setting, where the methods section can be put in the appendix. However, in conferences like ICML, the norm is to try and have everything needed to understand the paper in the main text. I would appreciate if some effort was made to make it like that.
    However, since this paper is fundamentally different from, say, papers that present new methods, I can let this slide.

2. Since details about methods are delegated to the Appendix, I find it quite necessary for the appropriate appendix section to be referenced in the main text, wherever it makes sense. This would make the paper much easier to read. However, the paper in its current form almost never refers to the corresponding appendix section in the main text.

---

> ### Author Rebuttal · Authors · 2026-03-30
>
> We are greatly encouraged by your support of our work!
>
> **W1**\
> We appreciate your understanding and we will extract subsection 3.1 and the first two paragraphs of subsection 3.2 from the Results section to create a new, dedicated Section 3: Models, datasets, and metrics.
>
> **W2**\
> We appreciate this constructive feedback and agree that cross-referencing will make the paper much easier to navigate. We will add references to: Appendices B.1–B.4 when introducing the auditory pre-trained models, B.5–B.6 when introducing the visual pre-trained models, and B.7–B.8 when detailing the evaluation metrics.
>
> **Q1**\
> The strong performance of supervised models like CochCNN9 and ResNet50 is largely expected given their training regime. These models were trained with supervision on English speech tasks. Because two fMRI datasets (NH2015 and B2021) consist of only native English speakers whose auditory cortices are highly tuned to the statistics of the English language, these supervised models benefit from a profound "in-distribution" advantage. Their high predictivity naturally reflects the alignment between the models' supervised linguistic targets and the human subjects' lifelong native-language expertise.
>
> Furthermore, the ability of these speech-trained models to predict auditory responses to 165 natural sounds highlights the robustness of their learned features. Because the CochCNN9 and ResNet50 models in our benchmark were optimized across three distinct supervised tasks (Word, Speaker, and Audio Event recognition), they were forced to extract rich, generalizable spectrotemporal modulations. These intermediate auditory representations happen to overlap with the statistics of environmental sounds. This shared statistical structure allows the supervised models to maintain strong predictive performance even when generalizing from speech to natural sounds.
>
> However, acknowledging the strong performance of these models actually reinforces our central claim. Despite CochCNN9 and ResNet50 possessing explicit English labels and multi-task supervision, our SSL models still achieve superior alignment of brain activity. Strikingly, they do so even when trained on entirely out-of-distribution stimuli (Mandarin). This contrast underscores our conclusion: adapting unsupervised learning to the naturalistic statistics of real-world data is a more potent and biologically plausible driver of brain-like representations than explicit, curated task supervision.
>
> **Q2**\
> The reviewer is correct that three ResNeXt models in Figure 8 are supervised models trained on ImageNet-1K. Our rationale for selecting these specific models was to control the architecture, as our baby-vision models also utilize the ResNeXt backbone. Following the reviewer's suggestion, we queried the Brain-Score benchmark and found three DINO models: vit_base_patch14_reg4_dinov2-lvd142m, dinov2-lvd142m, and dinov3-lvd1689m.
>
> While these models align with ours in utilizing an SSL objective, their training datasets create a fundamental discrepancy in test distribution on the Brain-Score benchmark. The fMRI benchmark stimuli consist of highly curated, adult-centric images. Because the DINO models were optimized on 142 million to 1.6 billion similarly curated, diverse web images (Oquab et al., 2023; Siméoni et al., 2025), they evaluate these benchmark stimuli In-Distribution (ID).
>
> In contrast, our baby-vision models are trained on egocentric footage bounded by the developmental and physical constraints of an infant's daily life. This biologically plausible data possesses a highly skewed, long-tailed distribution dominated by uncurated, often distorted close-ups of a limited set of household objects and caregivers. Canonical photos of distant vehicles are entirely foreign to this regime. Consequently, our baby-vision models are forced to predict brain responses to stimuli that are Out-of-Distribution (OOD).
>
> This dynamic perfectly mirrors our auditory findings. Just as native English speech is a "foreign," OOD stimulus to our models pretrained exclusively on Mandarin, curated ImageNet-style photographs are "foreign," OOD stimuli to our baby-vision models.
> With this context, the performance of vision models becomes striking. Despite a disadvantage in data volume, an older architecture, and the penalty of evaluating on OOD stimuli, our SSL baby-vision models achieve Brain-Scores (0.249 to 0.259) that are comparable with the ID-evaluated DINO SSL models (0.251 to 0.262).
>
> Together, our audio and vision results converge on a unified conclusion: SSL models trained on the statistics of naturalistic environments (whether baby-vision or foreign speech) learn representations so robust and universal that they can generalize to OOD stimuli as effectively as massive, brute-force models predicting ID stimuli.
>
> **Q3**\
> The correct number is indeed 8. While the original NH2015 dataset contains 10 participants, two participants only completed 2 scans and were therefore excluded.

---

> > ### Author Rebuttal · Reviewer_u4ti · 2026-04-06
> >
> > All questions have been answered adequately, except Q2:
> >
> > It is great to see that SSL baby-vision's brain-score is similar to Dinov2/3 despite it being trained on a lot more diverse data.
> >
> > I also apologize for the confusion from the lack of specificity in my question. My intention was to ask for DINOv1 as an example, which is pretrained through SSL on ImageNet1k alone. It would be great if the authors could add an ImageNet1k+SSL baseline (does not have to be DINOv1) as the fair comparison. This is so that the claim in the Fig 8 caption can be verified. It says "Unsupervised models trained on real-world data outperform ImageNet-pretrained models," but currently what is shown is only that _SSL models_ trained on real-world data output _supervised_ ImageNet-pretrained models.

---

> > > ### Author Response · Authors · 2026-04-06
> > >
> > > We thank the reviewer for catching this confounding variable! Following your suggestion, we evaluated an SSL + ImageNet1K baseline. Because a DINOv1 model was not natively available in the Brain-Score benchmark, we explicitly integrated and evaluated the DINO-ResNet50 model (pretrained via SSL on ImageNet1K using torch.hub.load('facebookresearch/dino:main', 'dino_resnet50')).
> > >
> > > * SSL + ID ImageNet1K (DINO-ResNet50): Brain-Score of **0.2051**
> > >
> > > * SSL + ID Massive Diverse Data (DINOv2/v3): Brain-Scores ranging from 0.251 to 0.262
> > >
> > > * SSL + OOD Real-World Data (baby-vision): Brain-Scores ranging from **0.249 to 0.259**
> > >
> > > By holding the unsupervised objective constant, this new baseline clearly demonstrates that the performance jump is indeed driven by the naturalistic, real-world OOD stimuli rather than just the SSL objective itself.
> > >
> > > We have uploaded these new results and the customized model evaluation code to our anonymous repository (https://anonymous.4open.science/r/21034).
> > >
> > > We are grateful for your constructive feedback throughout this process. We hope this fully resolves your final question!

---

### Official Review · Reviewer_j29Q · 2026-03-12

**Soundness:** 3
**Presentation:** 3
**Significance:** 2
**Originality:** 3
**Overall Recommendation:** 5
**Confidence:** 5

**Summary:**

Models that have been pretrained on real-world data, rather than curated data, better predict brain activity. This holds even for models that have been trained on out-of-distribution data (Mandarin) and tested on in-distribution data (English speakers’ brain activity). They do this for audio and vision models. They find that self-supervised pretrained audio models predict brain activity better. They control for dataset size by training on a small dataset and a large dataset of naturalistic sounds. They control for architecture by running an experiment where architecture is kept, but the training dataset is varied.

**Compliance With Llm Reviewing Policy:**

Affirmed.

**Final Justification:**

I think this paper is sound. I'm not sure if it is a very significant result, especially for the machine learning community. The rebuttal addressed my concerns about validity to large satisfaction.

**Key Questions For Authors:**

- Please elaborate on the contents and tasks of NH2015 and B2021 in 3.2. What are natural sounds?
- What exactly makes WenetSpeech more naturalistic than LibreSpeech? They're both speech datasets?
- Did I miss something, or are there any plans to do RDM analysis for the vision domain? This is not a major weakness, but it seems like something that would be good to do for completion.
- Would it be worth doing an analysis of which regions of the brain give unsupervised-models/realistic-data an edge? Or is this not feasible, since ROIs have already been pre-selected?

**Limitations:**

yes

**Strengths And Weaknesses:**

Strengths
- The paper is well written.
- Experiments are well structured: dataset size and architecture are controlled for.
- The result that unsupervised models result in better alignment with the brain is useful for the field to know.

Weaknesses
- I have reservations about the significance of the results. Is it really the case that something novel about the brain is being learned? The study evaluates the use of a dataset of patients listening to naturalistic sounds. So it seems like a more natural interpretation of the results would be: when brain alignment is evaluated using patients who listen to naturalistic sound, the highest alignment is found with models that have been trained on naturalistic sound. And perhaps this claim would hold if "naturalistic sound" were replaced with any other arbitrary acoustic property, e.g., "distorted sound". I think one experiment that would help clear things up would be this: can we see what the alignment is between the new models on a "clean" dataset? I argue that the alignment with the brain on this set will decrease.
- A common baseline that people do is alignment with an uninitialized model. It would be good to include that for completion. Is this what the SpectroTemporal baseline is?
- Can you make the comparison between figure 4 and figure 11 quantitative? Perhaps you can compare RDMs using Centralized Kernel Alignment? Mutual k-Nearest Neighbors?

---

> ### Author Rebuttal · Authors · 2026-03-31
>
> We sincerely thank the reviewer for their encouraging feedback!\
> Below is an anonymous link to a PDF containing Rebuttal figures and captions:\
> https://anonymous.4open.science/r/21034/Rebuttal.pdf
>
> **W1**\
> Please refer to our response to Reviewer iMjN (Question 1) and Rebuttal Figures 1–4, where we detail how our results capture two specific aspects of biological plausibility: cortical hierarchy and environmental adaptation.
>
> Beyond these current findings, a core contribution of our NeuroAI framework is its ability to generate testable hypotheses that guide future experiments. We strongly agree that the reviewer’s proposed experiment (evaluating model-brain alignment using fMRI data collected while subjects listen to explicitly "clean" versus "distorted" sounds) would be an elegant way to further isolate these effects. Unfortunately, the two existing fMRI datasets utilized in our study do not contain this specific stimulus contrast.
>
> **W2**\
> The SpectroTemporal baseline is the gray line shown in Figs. 2, 3, 9, and 10.
>
> We analyzed model unit activations in an uninitialized model (Rebuttal Figs. 6 and 7). We found that unit activation amplitudes and standard deviations are very similar across sounds, the tuning curves of model units are unrelated to fMRI voxels and components, and unit activations fail to explain component and voxel responses.
>
> **W3**\
> We thank the reviewer for suggesting these two methods to quantitatively compare the RDMs in Figures 4 and 11, an important addition also raised by Reviewer ps8P.
>
> Centered Kernel Alignment (CKA) is a robust metric used to evaluate global representational similarity (Kornblith et al., 2019). We applied CKA to the RDMs by double-centering the distance matrices and calculating their normalized Hilbert-Schmidt Independence Criterion (HSIC). On the NH2015 dataset, HuBERTspeech achieved a higher CKA score (0.7182) compared to ResNet50multitask (0.7012). This superior global alignment was replicated on the B2021 dataset (HuBERTspeech: 0.7331 vs. ResNet50multitask: 0.7013).
>
> Rebuttal Fig. 8 shows the quantitative comparison of RDMs using Mutual k-Nearest Neighbors (MkNN), which evaluates local topological geometry and neighborhood preservation. At small neighborhood sizes ($k < 40$ for NH2015; $k < 10$ for B2021), ResNet50multitask shares a higher proportion of immediate neighbors with the neural data, indicating stronger alignment with fine-grained, low-level acoustic clustering. Conversely, as the neighborhood expands ($k > 40$ for NH2015; $k > 10$ for B2021), the curves intersect, and HuBERTspeech demonstrates greater overlap.
>
> In summary, ResNet50multitask better captures fine-grained acoustic neighborhoods (small $k$ in MkNN), whereas HuBERTspeech better captures the overarching global geometry and broad categorical structure (large $k$ in MkNN and higher CKA).
>
> **Q1**\
> We will revise the first paragraph of Section 3.2 by adding a description of the tasks: "To ensure consistent engagement, both datasets required participants to perform a sound intensity discrimination task during scanning, pressing a button upon detecting a quieter target sound embedded within the stimulus blocks" and the definition of natural sounds: "These stimuli comprise 158 everyday sounds, rigorously screened via behavioral experiments for high human recognizability and daily exposure frequency, plus seven foreign speech clips".
>
> **Q2**\
> While both are indeed speech datasets, what makes a dataset "naturalistic" is an important question, which was also raised by Reviewer iMjN. To address this, we systematically quantified the audio quality and diversity differences between the LibriSpeech and WenetSpeech datasets in Rebuttal Fig. 5. We applied three deep learning models pretrained on either human perceptual evaluations or voice and speech datasets. We also measured a low-level audio metric.
>
> Together, we demonstrate a highly variable distribution of high-level perceptual, speech-specific, and low-level audio metrics among the 10 categories in WenetSpeech, confirming that it captures highly complex and naturalistic acoustic environments compared to the controlled recordings of LibriSpeech.
>
> **Q3**\
> We did not include an RDM analysis because neither of the referenced vision studies made their datasets publicly available. However, we definitely plan to add this RDM analysis if the original authors agree to share their stimulus images.
>
> **Q4**\
> It is indeed worth looking at the alignment across brain regions. We found that it is the non-primary lateral area, rather than the primary, anterior, or posterior areas, that gives SSL models trained on realistic data an edge. Please refer to Rebuttal Figures 1, 3, and 4 for these details. This also aligns with previous experimental studies by Kell and McDermott (2019), which showed that the non-primary auditory cortex located in the lateral, but not the anterior or posterior areas, exhibits noise-invariant responses to natural sounds.

---

> > ### Author Rebuttal · Reviewer_j29Q · 2026-04-03
> >
> > I thank the authors for taking the time to answer my questions. On point W1, the intention was to ask for a quick number: how do the models trained on "naturalistic data" perform on a "clean sound dataset". That is, can you benchmark your models' alignment with something like LibreSpeech? Does alignment go down or up after naturalistic training?

---

> > > ### Author Response · Authors · 2026-04-03
> > >
> > > We thank the reviewer for clarifying this point! We now understand that your intention was to ask about standard acoustic performance benchmarks on a "clean" dataset to see if naturalistic pretraining degrades performance.
> > >
> > > Intuitively, from a standard machine learning perspective, one might expect that a model trained on noisy, naturalistic data would perform worse on a clean dataset than a model explicitly trained on clean data. However, drawing on established benchmarks, we find that performance actually does not go down; it remains highly robust.
> > >
> > > To provide the specific numbers you requested using a clean, human speech benchmark (Speech Commands, Warden 2018; https://arxiv.org/pdf/1804.03209):
> > >
> > > * **Model trained on clean data:** The standard HuBERT Base model, pretrained on the highly curated, clean 960-hour LibriSpeech dataset, achieves an accuracy of **96.3%** on Speech Commands under the SUPERB benchmark (Yang et al., 2021; https://arxiv.org/pdf/2105.01051; https://huggingface.co/superb/hubert-base-superb-ks). *For exact reference, see Table 2; row: HuBERT Base; column: Keyword Spotting (KS).*
> > >
> > > * **Model trained on naturalistic data:** An adapted HuBERT model (AVES; Hagiwara et al., 2022; https://arxiv.org/pdf/2210.14493) pretrained on 360 hours of highly uncurated, real-world naturalistic and animal sounds (building directly upon our HuBERTcore distribution), achieves an accuracy of **96.4%** on the exact same clean Speech Commands dataset. *For exact reference, see Table 2; row: AVES-bio; column: sc.*
> > >
> > > This finding directly answers your original, foundational question: *"Is something novel about the brain being learned?"*\
> > > In standard engineering, shifting a model's training distribution from clean to noisy usually forces a severe trade-off. However, the human auditory cortex evolved in highly complex, noisy, naturalistic environments to be universally robust: our hearing does not suddenly degrade when we enter a quiet, clean room.
> > >
> > > The fact that self-supervised models pretrained on naturalistic data (1) achieve significantly higher alignment with the human brain (as shown in our fMRI results) and (2) maintain highly robust performance on clean acoustic datasets (96.4% vs 96.3%) demonstrates that these models are **successfully capturing this biologically inherent robustness.**
> > >
> > > We sincerely appreciate you pushing us on this point. Demonstrating that naturalistic training yields models that are both more brain-like and remarkably robust to clean environments greatly strengthens the significance of our NeuroAI framework.
> > >
> > > As this is the final interaction phase, we kindly ask that you update your overall assessment of our work in light of these clarifications.

---

### Official Review · Reviewer_iMjN · 2026-03-14

**Soundness:** 3
**Presentation:** 3
**Significance:** 3
**Originality:** 2
**Overall Recommendation:** 4
**Confidence:** 4

**Summary:**

This paper looks into how factors such as architecture, training objective, and data distribution influence the alignment between artificial neural networks and biological neural activity in human visual and auditory cortices.

A comparison is made between supervised and unsupervised models trained on curated datasets vs. naturalistic datasets. Two main metrics are used for evaluation: encoding models (voxelwise regression for predicting fMRI responses) and representation similarity analysis (RSA).

Results revealed that unsupservised models trained on richer, realistic data were better at predicting neural responses including those to out-of-distribution examples (e.g., models trained on spoken Mandarin predicting brain responses to spoken English). The authors argue that the training data distribution is the main factor providing the alignment with the brain, rather than the architecture or dataset size.

**Compliance With Llm Reviewing Policy:**

Affirmed.

**Final Justification:**

I am satisfied with the responses from the authors to my concerns. They have also provided their code with detailed use instructions, which ensures reproducibility and increases the Soundness of the submission. Therefore, I am happy to increasing my score. I still think this study offers a minor contribution to the field, so I am giving it a 4.

**Key Questions For Authors:**

(i) May the authors clarify in what way do the results could imply biological plausibility beyond representational alignment or predictive capacity? How to distinguish ,in this context, between statistical alignment and mechanistic correspondence?

(ii) The paper suggests that the training data distribution is a key factor for improving neural response prediction. Could the authors discuss in more detail which properties of the naturalistic datasets (e.g. noise, long-tail, redundancy) are chiefly responsible for this effect? E.g., does introducing any of these characteristics into the curated datasets lead to similar improvement in the performance of the resulting models? That would be a way to objectively test the proposed explanations.

(iii) I found no specific details about how the new models were trained. Could the authors provide the necessary code (e.g., on Anonymous GitHub) to reproduce this study and achieve the same results?

**Limitations:**

The paper does not clearly state the limitations of the study. As one example, the authors could expand the Discussion by explicitly mentioning the limitations of the evaluation used and clarifying that better performance in neural prediction metrics do not necessarily imply a mechanistic correspondence with the brain.

**Strengths And Weaknesses:**

__Strengths:__

The paper is well-written and clear, and discusses a central problem in neuroAI: which aspects of training can lead to models that are better-aligned to biological neural data.
The idea of comparing supervised vs. unsupervised settings together with curated vs. naturalistic data and in two sensory domains provides for a quite interesting analysis.
The authors use two different metrics and both reach similar results.

__Weaknesses:__

__Soundness:__
(i)
The paper interprets improvements in neural prediction metrics as gold standard evidence for biological plausibility. However, the metrics used (encoding models and RSA) are limited to representational or predictive alignment, disregarding mechanistic correspondence between models and the brain. E.g., in an encoding model good matching might mean that the model features contain enough information about the stimuli, so regression can reconstruct the neural responses accurately; it does not necessarily imply that the computational representations in the model are similar to those in the brain. As for RSA, it checks if stimuli that are similar to the brain are also similar to the model (via pairwise stimulus distances); similar representational geometry does not necessarily mean similar learned features, nor dynamics, so it's difficult to draw any conclusions about shared computational mechanisms. This is a clear limitation of the study and there is no discussion about this in the paper whatsoever.

A few suggestions that could make this analysis stronger: tuning curve comparisons between model units and neurons; adversarial manipulations; robustness to transformations. In fact, given the central argument of the paper about the impact of real-world statistics in forming neural representations, it would be interesting to test if the models learn properties of invariance and equivariance similar to those observed in real neural responses, for example by means of controlled stimulus transformations.

__Originality:__
(ii)
The authors mention in the Discussion that the finding that models trained in more naturalistic datasets can generalize better to novel stimuli may reflect a regularization mechanism. It has been previously shown that data augmentation techniques can have various effects on generalization (both ID and OOD) by means of implicit regularization, e.g.:

Lin, C. H., Kaushik, C., Dyer, E. L., & Muthukumar, V. (2024). The good, the bad and the ugly sides of data augmentation: An implicit spectral regularization perspective. Journal of Machine Learning Research, 25(91), 1-85.


(iii)
The fact that an unsupervised model trained on a richer, more realistic dataset can result in better generalization performance than a supervised one trained on impoverished, curated data is interesting. However, the fact that self-supervised networks learning from unlabeled data may outperform supervised ones is not new, especially for representation learning and transfer learning (one classic example is in CLIP-style pretraining).

---

> ### Author Rebuttal · Authors · 2026-03-30
>
> We sincerely thank the reviewer for their encouraging feedback!\
> Below is an anonymous link to a PDF containing Rebuttal figures and captions:\
> https://anonymous.4open.science/r/21034/Rebuttal.pdf
>
> **Soundness**\
> We thank the reviewer for highlighting the limitations of these two metrics. Per your and Reviewer j29Q’s suggestions, we will add a new paragraph:\
> *Note that the two metrics used are limited to predictive or representational alignment, disregarding mechanistic correspondence between models and the brain. In an encoding model, good matching might mean that the model features contain enough information about the stimuli. In RSA, similar representational geometry does not necessarily mean similar learned features. To complement them, we take two strategies. One is to compare the tuning curves of the model units with the fMRI voxel or component. The other is to compare RDMs between model activations and brain responses using Centered Kernel Alignment (CKA) and Mutual k-Nearest Neighbors (MkNN).*
>
> We compared tuning curves between model units and fMRI voxels or decomposed components.\
> In Rebuttal Fig. 1, we compared model unit against fMRI voxel in the lateral auditory cortex. Speech and vocal music are clearly distinguished from other categories in both heatmaps. Furthermore, model unit tuning curves match fMRI voxel tuning curves.\
> In Rebuttal Fig. 2, we compared model units with decomposed fMRI speech components. Example speech and mechanical sounds evoke unique unit activation patterns, showing larger amplitudes and higher fluctuations for mechanical sounds than speech. Importantly, their corresponding component responses are also well-separated, exhibiting a significant negative correlation with the standard deviation of model activations. Notably, these relationships disappear in permuted models (Rebuttal Figs. 6 and 7).\
> Therefore, although units and voxels are physically distinct and their activation signs are reversed, we conclude that their tuning curves to natural sounds closely match.
>
> **Originality**\
> We emphasize that the primary scope of our study is NeuroAI, not general machine learning (ML). Our originality lies in demonstrating that well-established ML principles also benefit models of brain. While data augmentation and SSL are known to improve ML performance, their specific impact on model-brain alignment was previously unclear. Demonstrating that these techniques successfully bridge ML and biological learning highlights our core contribution, rather than compromising it.
>
> To properly contextualize our findings within the broader ML literature, we will add the following text:\
> *Our findings are consistent with modern machine learning literature (Lin et al., 2024), where both ID and OOD data augmentation improve generalization via implicit regularization.*
>
> **Q1**\
> Our results highlight two aspects of biological plausibility. First, regarding cortical hierarchy: the tuning curve consistency between model units and voxels/components (Rebuttal Figs. 1 and 2) mainly holds true in the Lateral auditory cortex, but not in the Anterior, Posterior, and Primary areas (Rebuttal Fig. 3). Second, reflecting the brain's adaptation to real-world environments: pretraining on naturalistic sounds (HuBERTspeech) increases explained variance compared to pretraining on curated sounds (HuBERTLS) across both datasets and all four auditory cortical areas (Rebuttal Fig. 4).
>
> **Q2**\
> Our current results already show that introducing properties of naturalistic datasets improves model performance. As shown in Figs. 2 and 3, the HuBERTcore SSL model outperforms three other SSL models: Wav2Vec2LS, Wav2Vec2FT, and HuBERTLS. HuBERTcore was pretrained on a smaller (153 hours) but more diverse dataset, compared to the much larger (960 hours), audiobook-only LibriSpeech (LS) dataset. Therefore, by comparing models that share the same architecture and objective, but differ in their training data, our argument regarding the critical role of training data distribution largely holds.
>
> **Q3**\
> We used pretrained models hosted on HuggingFace to extract model activations.
> We will add the following sentences to the end of Appendix B.1:\
> *Details on the models' original training process can be found at: https://github.com/TencentGameMate/chinese_speech_pretrain. As per the developers, the BASE model was trained using 8 A100 GPUs, and the LARGE model used 16 A100 GPUs.*\
> Rebuttal constraints only allow us to share links for Figures and captions. We will make all code publicly available upon acceptance.
>
> **Limitations**\
> We will add:\
> *One limitation of our study is the evaluation metrics we used, including the encoding model, RSA, CKA, and MkNN, where better performance in those metrics does not necessarily imply a mechanistic correspondence with the brain. Although comparing tuning curves between model units and fMRI voxels or components complements those metrics, more analysis remains to be done in the future.*

---

> > ### Author Rebuttal · Reviewer_iMjN · 2026-04-02
> >
> > Thanks for the detailed responses. The authors have mainly addressed my points of concern, except for sharing the code required to reproduce their results. Using the same link to https://anonymous.4open.science/r/21034/, you can easily share more files to include your code and instructions for how to run it to generate your results.

---

> > > ### Author Response · Authors · 2026-04-03
> > >
> > > We sincerely thank the reviewer for confirming that our previous responses successfully addressed your **main concerns**.
> > >
> > > We agree with the importance of reproducibility and appreciate you pushing us to provide these resources during the discussion phase.
> > >
> > > Following your instruction, we have fully updated the anonymous repository to include all code and instructions required to reproduce our results.
> > >
> > > Specifically, we have added:
> > >
> > > *Full Source Code & Notebooks:*\
> > > All scripts used to generate the results, alongside Jupyter Notebooks containing the code to visualize the figures.
> > >
> > > *Model Activations:*\
> > > The code used to extract model unit activations from our five customized models, as well as the extracted activation files themselves.
> > >
> > > *Comprehensive README:*\
> > > A detailed, step-by-step README.md file (https://anonymous.4open.science/r/21034/README.md) that provides run instructions and explicitly maps our scripts to every specific figure in both the main manuscript and the rebuttal.
> > >
> > > Since code availability was your **final concern**, we hope this comprehensive update fully resolves it.
> > >
> > > As this is the final interaction phase, we kindly ask that you reconsider your overall assessment of our work in light of these additions.

---

### Decision · Program_Chairs · 2026-04-30

**Decision:**

Accept (spotlight)

**Comment:**

This article, among other intriguing findings, shows that SSL models trained on real-world data outperform supervised models in predicting visual and auditory brain responses. The authors and reviewers engaged in a lively and productive rebuttal discussion, during which the authors provided several follow-up clarifications, plots and promised to include improvements into their manuscript (including new paragraphs, the release of code, etc). Overall, reviewers asses this to be a solid and well-written submission, with interesting findings for the NeuroAI community.